# Repelling Random Walks

**Isaac Reid**[1] , **Eli Berger**[2] , **Krzysztof Choromanski**[3,4]*, **Adrian Weller**[1,5]
[1]University of Cambridge, [2]University of Haifa, [3]Google DeepMind,
[4]Columbia University, [5]Alan Turing Institute
`ir337@cam.ac.uk`, `kchoro@google.com`

### Abstract

We present a novel quasi-Monte Carlo mechanism to improve graph-based sampling, coined *repelling random walks*. By inducing correlations between the trajectories of an interacting ensemble such that their marginal transition probabilities are unmodified, we are able to explore the graph more efficiently, improving the concentration of statistical estimators whilst leaving them unbiased. The mechanism has a trivial drop-in implementation. We showcase the effectiveness of repelling random walks in a range of settings including estimation of graph kernels, the PageRank vector and graphlet concentrations. We provide detailed experimental evaluation and robust theoretical guarantees. To our knowledge, repelling random walks constitute the first rigorously studied quasi-Monte Carlo scheme correlating the directions of walkers on a graph, inviting new research in this exciting nascent domain.[1]

## 1 Introduction and related work

Quasi-Monte Carlo (QMC) sampling is well-established as a universal tool to improve the convergence of MC methods, improving the concentration properties of estimators by using low-discrepancy samples to reduce integration error (Dick et al., 2013). They replace i.i.d. samples with a correlated ensemble, carefully constructed to be more 'diverse' and hence improve approximation quality.

Such methods have been widely adopted in the Euclidean setting. For example, when sampling from isotropic distributions, one popular approach is to condition that samples are orthogonal: a trick that has proved successful in applications including dimensionality reduction (Choromanski et al., 2017), evolution strategy methods in reinforcement learning (Choromanski et al., 2018; Rowland et al., 2018) and estimating sliced Wasserstein distances (Rowland et al., 2019). 'Orthogonal Monte Carlo' has also been used to improve the convergence of random feature maps for kernel approximation (Yu et al., 2016), including recently in attention approximation for scalable Transformers (Choromanski et al., 2020). Intuitively, conditioning that samples are orthogonal prevents them from clustering together and ensures that they 'explore' $\mathbb{R}^d$ better. In specific applications it is sometimes possible to derive rigorous theoretical guarantees (Reid et al., 2023b).

Less clear is how these powerful ideas generalise to discrete space. Of particular interest are *random walks on graphs*, which sample a sequence of nodes connected by edges with some stopping criterion. Random walks are ubiquitous in machine learning and statistics (Xia et al., 2019), providing a simple mechanism for unbiased graph sampling that can be implemented in a distributed way. However, slow diffusion times (especially for challenging graph topologies) can lead to poor convergence and downstream performance.

Our key contribution is the first (to our knowledge) quasi-Monte Carlo scheme that correlates the *directions* of an ensemble of graph random walkers to improve estimator accuracy. By conditioning that walkers 'repel' in a particular way that leaves the marginal walk probabilities unmodified, we are able to provably suppress the variance of various estimators

---

*Senior lead.
[1]Code is available at https://github.com/isaac-reid/repelling_random_walks.

whilst preserving their unbiasedness. We derive strong theoretical guarantees and observe large performance gains for algorithms estimating three disparate quantities: graph kernels (Choromanski, 2023), the PageRank vector (Avrachenkov et al., 2007) and graphlet concentrations (Chen et al., 2016).

**Related work:** The poor mixing of random walkers on graphs is well-documented and various schemes exist to try to improve estimator convergence. Most directly modify the base Markov chain by changing the transition probabilities, but without altering the walker's stationary distribution and therefore leaving *asymptotic* estimators (e.g. based on empirical node occupations) unmodified. The canonical example of such a scheme is *non-backtracking walks* which do not permit walkers to return to their most recently visited node (Alon et al., 2007; Diaconis et al., 2000; Lee et al., 2012). More involved schemes allow walkers to interact with their entire history (Zhou et al., 2015; Doshi et al., 2023). Many of these strategies provide theoretical guarantees that the asymptotic variance of estimators is reduced, but crucially the *marginal* probabilities of sampling different walks are modified so they cannot be applied to non-asymptotic estimators that rely on particular known marginal transition probabilities. Conversely, our QMC scheme leaves marginal walk probabilities unmodifed. Research has also predominantly been restricted to the behaviour of a *single self-interacting walker* rather than an ensemble, and when multiple walkers are considered analytic results are generally restricted to simple structures, e.g. complete graphs (Rosales et al., 2022; Chen, 2014). This research exists within the broader literature of *reinforced random walks*, where nonlinear Markov kernels are used so that walkers are less (or more) likely to transition to nodes that have been visited in the past (Pemantle, 2007). However, the analytic focus has predominantly been on properties like recurrence times, escape times from sets, cover times and localisation results for simple topologies (Amit et al., 1983; Tóth, 1995; Tarrès, 2004), rather than the behaviour of associated statistical estimators on general graphs. The latter is of more direct interest in machine learning.

The remainder of the manuscript is organised as follows. In **Sec. 2** we introduce the requisite mathematics and present our novel QMC repelling random walk mechanism. In **Secs 3-5** we use it to approximate three quantities of interest in machine learning: graph node kernels (**Sec. 3**), the PageRank vector (**Sec. 4**), and graphlet statistics (**Sec. 5**). Repelling random walks are empirically found to outperform the i.i.d. variant in every case and we are often able to provide concrete theoretical guarantees.

## 2 REPELLING RANDOM WALKS

Consider an undirected, connected graph $\mathcal{G}(\mathcal{N}, \mathcal{E})$ where $\mathcal{N} := \{1, ..., N\}$ denotes the set of nodes and $\mathcal{E}$ denotes the set of edges, with $(i, j) \in \mathcal{E}$ if there is an edge between nodes $i, j \in \mathcal{N}$. Write the graph's (weighted) adjacency matrix $\mathbf{A} := [a_{ij}]_{i,j \in \mathcal{N}}$, where $a_{ij} \neq 0$ if $(i, j) \in \mathcal{E}$ and 0 otherwise. Let $d_i := \sum_{j \in \mathcal{N}} \mathbb{I}[(i, j) \in \mathcal{E}]$ denote the node degree, which is the number of neighbours of a particular node, and let $\mathcal{N}(i) := \{j \in \mathcal{N} | (i, j) \in \mathcal{E}\}$ denote the set of neighbours of node $i$. The transition matrix $\mathbf{P} = [P_{ij}]_{i,j \in \mathcal{N}}$ of a *simple random walk* is given by

$$P_{ij} = \begin{cases} \frac{1}{d_i} & \text{if } (i, j) \in \mathcal{E} \\ 0 & \text{otherwise} \end{cases} \tag{1}$$

such that at every timestep the walker selects one of its neighbours with uniform probability. This can be viewed as a finite and time-reversible Markov chain with state space $\mathcal{N}$.

Supposing we have $m$ such walkers on the graph simultaneously, we can define an *augmented Markov chain* with state space $\mathcal{N}^m$, consisting of the possible node positions of all the walkers. If the walkers are independent, the joint transition matrix $\mathbf{Q} \in \mathbb{R}^{N^m \times N^m}$ is given a Kronecker product of the marginal transition matrices $\mathbf{P}^{(i)}$:

$$\mathbf{Q} = \otimes_{i=1}^{m} \mathbf{P}^{(i)} \tag{2}$$

where the index $i = 1, ..., m$ enumerates the walkers present. Our key contribution is now to induce correlations between the walkers' paths such that the *joint* transition matrix $\mathbf{Q}$ is modified but each *marginal* transition matrix (and hence the unbiasedness of any estimator relying on it) is unchanged. The correlations are designed to improve estimator convergence.

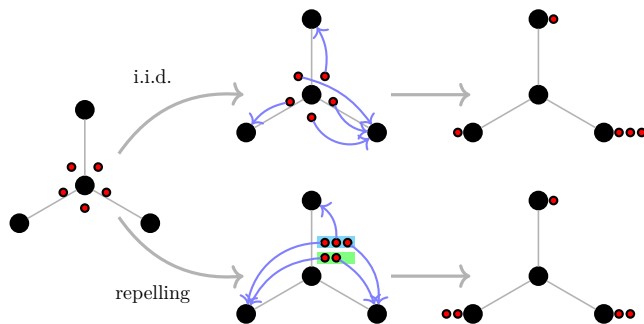

Figure 1: Schematic for behaviour of repelling random walkers at a particular timestep. By sampling from each 'block' (blue and green rectangles) without replacement we get a more even distribution over neighbours, without changing the marginal probabilities.

**Definition 2.1** (Repelling random walks). *A repelling* ensemble has the following behaviour. *Let $\mathcal{V}_t^{(i)}$ denote the set of walkers at node $i$ at timestep $t$, and $N_t^{(i)} := |\mathcal{V}_t^{(i)}|$ the size of this set. Randomly divide these walkers into $N_t^{(i)}//d$ subsets of size $d$ and one 'remainder' subset of size $N_t^{(i)}\%d < d$ (where $//$ and $\%$ denote truncating integer division and the remainder, respectively). Among each subset, assign the walkers to a neighbour from the set $\mathcal{N}(i)$ uniformly* without replacement.

This is in contrast to i.i.d. walkers where $\mathcal{V}_t^{(i)}$ are assigned to the neighbours $\mathcal{N}(i)$ uniformly *with replacement*. We provide a schematic in Fig. 1. In the repelling scheme, each walker still has a *marginal* transition probability $P_{ij} = \{1/d_i$ if $(i,j) \in \mathcal{E}$, 0 otherwise$\}$, but now they are forced to take different edges and heuristically 'explore' the graph more effectively. The sample of walks is more 'diverse'. Since the marginal transition probabilities $\mathbf{P}^{(i)}$ are unmodified, any estimators that are unbiased with i.i.d. walkers are also *automatically unbiased* with repelling walkers, including in the non-asymptotic regime. However, as we shall see, their concentration properties are often substantially better.

**Computational cost and implementation**: Repelling random walks have a trivial drop-in implementation. The only difference is whether walkers are assigned to neighbours with or without replacement. Moreover, the transitions in the augmented state space $\mathcal{N}^m$ remain Markovian (memoryless); there are no extra space complexity costs because we only need access to the current positions of all the walkers.

**Physical interpretation and entanglement:** Under repulsive interactions, the joint transition matrix $\mathbf{Q}$ can no longer be written as a Kronecker product. Consider transitions of 2 walkers in the same 'block' from $(i_1, i_2)$ to $(j_1, j_2)$. We have:

$$\mathbf{Q}_{Ni_1+i_2, Nj_1+j_2} := \Pr(j_1, j_2|i_1, i_2) = \mathbf{P}_{i_1j_1}\mathbf{P}_{i_2j_2} \cdot \begin{cases} 1 + \delta_{i_1i_2}\left(\frac{d_i}{d_i-1}(1-\delta_{j_1j_2})-1\right) & \text{if } d_i \neq 1 \\ 1 & \text{if } d_i = 1 \end{cases}$$

$$(3)$$

with $\delta_{i_1i_2}$ the delta function. This does not generically factorise into $(i_1, j_1)$- and $(i_2, j_2)$-dependent parts. In quantum mechanics (QM), an interacting Hamiltonian $H$ which cannot be written as a Kronecker *sum* gives rise to a time-evolution operator $U := \exp(-\frac{i}{\hbar}Ht)$ that cannot be written as a Kronecker *product*, which in turn generically gives rise to quantum entanglement between particles. Just as the von-Neumann entropy (a measure of bipartite quantum entanglement (Amico et al., 2008)) increases under such interactions, in our QMC scheme the Shannon mutual information initially increases from 0: during the first timestep, $\Delta I_{1,2} = \delta_{i_1i_2}\log(\frac{d_i}{d_i-1}) \geq 0$. Note that the analogy is not exact because in QM the time-evolution operator acts on (complex) wavefunctions whereas here the transition matrix acts on the (real positive) probabilities of being in different states of a Markov chain encoding the positions of walkers on a graph. It is just intended to help build intuition for the reader.

It will be convenient to define one further class of interacting random walk.

**Definition 2.2** (Transient repelling random walks). *An ensemble of random walks is described as* transient repelling *the walkers repel (according to Def. 2.1) at the first timestep, and are independent thereafter.*

Such an ensemble will capture the repelling behaviour at early times but eventually relax to independence. Whilst less practical than the full repelling scheme, we will see that sometimes it makes theoretical analysis tractable.

We now apply our repelling random walks mechanism to three disparate applications: approximation of graph kernels (Sec. 3), approximation of the PageRank vector (Sec. 4), and approximation of graphlet concentrations (Sec. 5).

## 3 APPLICATION 1: APPROXIMATING GRAPH KERNELS

We begin by demonstrating the effectiveness of repelling random walks for estimating *graph kernels* $K_{\mathcal{G}} : \mathcal{N} \times \mathcal{N} \to \mathbb{R}$, defined on the nodes $\mathcal{N}$ of a graph $\mathcal{G}$. Such kernels capture the structure of $\mathcal{G}$, letting practitioners repurpose theoretically grounded and empirically successful algorithms like support vector machines, kernelised principal component analysis and Gaussian processes to the discrete domain (Smola and Kondor, 2003). Applications include in bioinformatics (Borgwardt et al., 2005), community detection (Kloster and Gleich, 2014), generative modelling (Zhou et al., 2020) and solving shortest-path problems (Crane et al., 2017). Chief examples of $K_{\mathcal{G}}$ are the $d$-regularised Laplacian and diffusion kernels, given by $\mathbf{K}_{\text{lap}}^{(d)} \coloneqq (\mathbf{I} + \sigma^2 \widetilde{\mathbf{L}})^{-d}$ and $\mathbf{K}_{\text{diff}} \coloneqq \exp(-\sigma^2 \widetilde{\mathbf{L}}/2)$ respectively. Here, $\sigma^2$ is a lengthscale parameter and $\widetilde{\mathbf{L}}$ is the *normalised graph Laplacian*, defined by $\widetilde{\mathbf{L}} \coloneqq \mathbf{I} - \mathbf{W}$ with $\mathbf{W} = [a_{ij}/(\tilde{d}_i \tilde{d}_j)^{1/2}]_{i,j=1}^N$ a normalised weighted adjacency matrix ($\tilde{d}_i = \sum_j a_{ij}$ is the weighted node degree and $a_{ij}$ is the weight of the original edge). $\widetilde{\mathbf{L}}$ is the analogue of the familiar Laplacian operator $\nabla^2 = \frac{\partial^2}{\partial x_1^2} + \frac{\partial^2}{\partial x_2^2} + ... + \frac{\partial^2}{\partial x_n^2}$ in discrete space, describing diffusion on $\mathcal{G}$ (Chung and Yau, 1999; Chung, 1997).

For large graphs, computing e.g. $\mathbf{K}_{\text{lap}}^{(d)}$ exactly can be prohibitively expensive due to the $\mathcal{O}(N^3)$ time complexity of matrix inversion. This motivated the recently-introduced class of *Graph Random Features* (GRFs) (Choromanski, 2023), which provide a discrete analogue to Random Fourier Features (Rahimi and Recht, 2007). These $N$-dimensional vectors $\phi(i) \in \mathbb{R}^N$ are constructed for every node $i \in \mathcal{N}$ such that their Euclidean dot product is equal to the kernel evaluation in expectation,

$$[\mathbf{K}_{\text{lap}}^{(2)}]_{ij} = \mathbb{E}\left(\phi(i)^\top \phi(j)\right). \tag{4}$$

In their paper, Choromanski (2023) provides an elegant algorithm for constructing $\phi(i)$: one simulates $m \in \mathbb{N}$ random walks out of each node $i$ that terminate with probability $p$ at every timestep, depositing a 'load' at every node they visit to build up a randomised projection of the local environment in $\mathcal{G}$. They show that this gives an unbiased estimate of $\mathbf{K}_{\text{lap}}^{(2)}$, which can be used to construct $\mathbf{K}_{\text{lap}}^{(d)}$ for $d \in \mathbb{N}$ or an asymptotically unbiased approximation of $\mathbf{K}_{\text{diff}}$. Since the unbiasedness of the estimator depends on the marginal probabilities of sampling different finite-length random walks being unmodified (c.f. just its stationary distribution), it is a natural setting to test our new quasi-Monte Carlo scheme.

Remarkably, under mild conditions, we are able to derive an analytic closed form for the difference in kernel estimator mean squared error (MSE) between the i.i.d. and transient-repelling mechanisms for general graphs (deferred to Eq. 32 in App. A.1 for brevity). This enables us to make the following statement for some specific graphs, proved in App. A.1.

**Theorem 3.1** (Superiority of repelling random walks for kernel estimation). *Consider graph nodes indexed $(i, j)$ separated by at least 2 edges. In the limit $\sigma \to 0$, provided the number of walkers in the transient repelling ensemble is smaller than or equal to the node degrees $d_{\{i,j\}}$ and the edge-weights of $\mathbf{W}$ are equal,*

$$Var([\widehat{\mathbf{K}}_{lap\,ij}^{(2)}]_{repelling}) \le Var([\widehat{\mathbf{K}}_{lap\,ij}^{(2)}]_{i.i.d.}) \tag{5}$$

*for both i) trees and ii)* 2*-dimensional grids.*

Though we have made some restrictions for analytic tractability, we will empirically observe that the full repelling QMC scheme is effective in much broader settings. In particular,

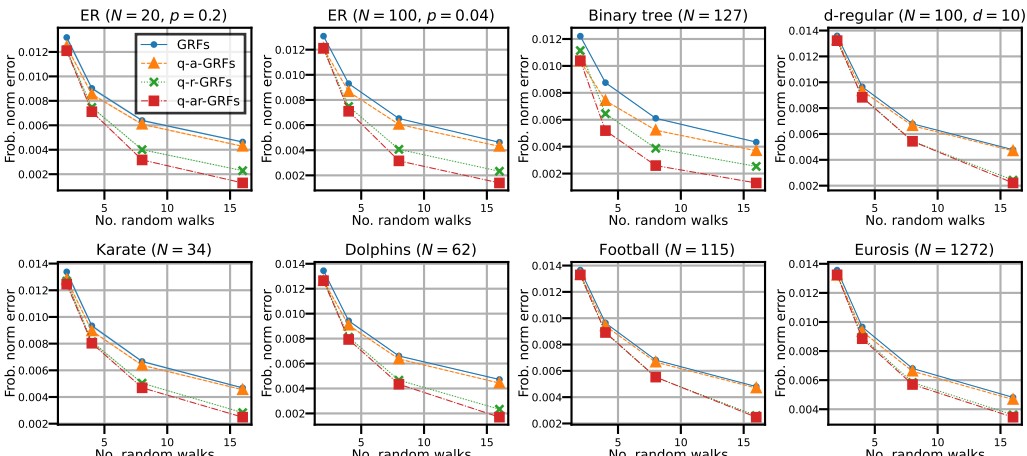

Figure 2: Relative Frobenius norm of estimates of the 2-regularised Laplace kernel (lower is better) vs. number of random walks for: i) vanilla GRFs; ii) GRFs with antithetic termination (Reid et al., 2023a) ('q-a-GRFs'); iii) GRFs with repelling walks ('q-r-GRFs'); iv) GRFs with both antithetic termination and repelling walks ('q-ar-GRFs'). Using both QMC schemes together gives the best results for all graphs considered and the gains are large (sometimes a factor of $> 2$). $N$ gives the number of nodes, $p$ is the edge-generation probability for the Erdös-Rényi graphs, and $d$ is the $d$-regular node degree. One standard deviation on the mean error is shaded but is too small to easily see.

it substantially suppresses kernel estimator variance with many walkers, arbitrary $\sigma$ and arbitrary graphs. Extending the proof to these general cases is an exciting open problem.

We also note that our scheme is fully compatible with the recently-introduced QMC scheme known as *antithetic termination* (Reid et al., 2023a), which anticorrelates the *lengths* of random walkers (by coupling their terminations) but does not modify their trajectories. Both schemes can be applied simultaneously, inducing negative correlations between both the walk directions and lengths.

### 3.1 Pointwise kernel estimation

We now empirically test Eq. 5 for general graphs by comparing the variance of $[\widehat{\mathbf{K}}^{(2)}_{\mathrm{lap}}]_{ij}$ under different schemes. In what follows, 'GRFs' refers to graph random features constructed using i.i.d. walkers, whilst 'q-{a,r,ar}-GRFs' denotes the efficient quasi-Monte Carlo variants where walkers exhibit antithetic termination ('a') (Reid et al., 2023a), repel ('r'), or both ('ar'). We use these different flavours of (q-)GRFs to generate unbiased estimates $\widehat{\mathbf{K}}^{(2)}_{\mathrm{lap}}$, then compute the relative Frobenius norm $\|\mathbf{K}^{(2)}_{\mathrm{lap}} - \widehat{\mathbf{K}}^{(2)}_{\mathrm{lap}}\|_{\mathrm{F}}/\|\mathbf{K}^{(2)}_{\mathrm{lap}}\|_{\mathrm{F}}$ between the true and approximated Gram matrices. Fig. 2 presents the results for various graphs: small Erdős-Rényi, larger Erdős-Rényi, a binary tree, a $d$-regular graph, and four standard real-world examples from (Ivashkin, 2023) (`karate`, `dolphins`, `football` and `eurosis`). These differ substantially in both size and structure. We take 100 repeats to compute the variance of the kernel approximation error, using a regulariser $\sigma = 0.1$ and a termination probability $p = 0.5$. The gains provided by the repelling QMC scheme (green) are much greater than those from antithetic termination (orange), but the lowest variance is achieved when both are used together (red). Note that the gains provided by repelling random walks continue to accrue as the size of the ensemble grows; with $m = 16$ walkers the error is often halved.

### 3.2 Downstream applications: kernel regression for node attribute prediction

We have both proved (Theorem 3.1) and empirically confirmed (Fig. 2) that using repelling random walks substantially improves the quality of estimation of the 2-regularised Laplacian

kernel using GRFs. Naturally, this permits better performance in downstream applications that depend on the approximation. As an example, we follow Reid et al. (2023a) and consider kernel regression on triangular mesh graphs (Dawson-Haggerty, 2023).

Consider a graph $\mathcal{G}$ where each node is associated with a normal vector $\boldsymbol{v}^{(i)}$. The task is to predict the directions of a random set of missing 'test' vectors (a 5% split) from the remaining 'train' vectors. We compute our (unnormalised) predictions $\widehat{\boldsymbol{v}}^{(i)}$ as $\widehat{\boldsymbol{v}}^{(i)} \coloneqq \sum_j \widehat{\mathbf{K}}_{\mathrm{lap}}^{(2)}(i,j)\boldsymbol{v}^{(j)}$, where $j$ sums over the training vertices and $\widehat{\mathbf{K}}_{\mathrm{lap}}^{(2)}(i,j)$ is constructed using the GRF and q-{a,r,ar}-GRF mechanisms described in Sec. 3.1. We compute the average angular error $1 - \cos\theta$ between the prediction $\widehat{\boldsymbol{v}}^{(i)}$ and groundtruth $\boldsymbol{v}^{(i)}$ across the test set. We use $m = 16$ random walks with a termination probability $p = 0.5$ and a regulariser $\sigma = 0.1$, taking 1000 repeats for statistics. Table 1 reports the results. Higher-quality kernel approximations with repelling random walks give more accurate downstream predictions for all graphs, with the biggest gains appearing when our repelling scheme is introduced ('r' and 'ar'). The difference is remarkably big when the number of nodes $N$ is big: on `torus`, the error is reduced by a factor of almost 3. Accurate approximation is especially helpful for these large graphs as exact methods become increasingly expensive.

Table 1: Angular error $1 - \cos\theta$ between true and predicted node vectors when approximating the Gram matrix with GRFs and q-{a,r,ar}-GRFs (lower is better). Brackets give one standard deviation. Both schemes in combination works best.

| Graph | $N$ | Pred error, $1 - \cos\theta$ | | | |
|---|---|---|---|---|---|
| | | GRFs | q-a-GRFs | q-r-GRFs | q-ar-GRFs |
| cylinder | 210 | 0.0650(7) | 0.0644(7) | 0.0466(3) | **0.0459(2)** |
| teapot | 480 | 0.0331(2) | 0.0322(2) | 0.0224(1) | **0.0215(1)** |
| idler-riser | 782 | 0.0528(3) | 0.0521(3) | **0.0408(2)** | **0.0408(2)** |
| busted | 1941 | 0.00463(2) | 0.00456(2) | 0.003833(6) | **0.003817(6)** |
| torus | 4350 | 0.000506(1) | 0.000482(1) | **0.000180(1)** | **0.000181(1)** |

Though for concreteness we have considered one particular downstream application, we stress that improving the kernel estimate can be expected to boost performance in any algorithm that uses it, e.g. for graph node clustering (Dhillon et al., 2004), shortest-path prediction (Crane et al., 2017) or simulation of graph diffusion (Reid et al., 2023a).

## 4 APPLICATION 2: APPROXIMATING PAGERANK

As a second application, we use repelling random walks to improve numerical estimates of the *PageRank* vector: a popular measure of node importance in a graph originally proposed by Page et al. (1998) to rank websites in search engine results. The PageRank vector is defined as the stationary distribution of Markov chain whose state space is the set of all graph nodes $\mathcal{N}$, with a transition matrix

$$\widetilde{\mathbf{P}} \coloneqq (1-p)\mathbf{P} + \frac{p}{N}\mathbf{E}. \qquad (6)$$

Here, $p \in (0,1)$ is a scalar, $N$ is the number of nodes, $\mathbf{P}$ is defined in Eq. 1 and $\mathbf{E} = [1]_{i,j \in \mathcal{N}}$ is a matrix whose entries are all ones. This encodes the behaviour of a 'surfer' who at every timestep either teleports to a random node with probability $p$ or moves to one of its neighbours chosen uniformly at random. Since $\widetilde{\mathbf{P}}$ is stochastic, aperiodic and irreducible, we can define the unique PageRank vector $\boldsymbol{\pi} \in \mathbb{R}^N$:

$$\boldsymbol{\pi}^\top \widetilde{\mathbf{P}} = \boldsymbol{\pi}^\top, \quad \boldsymbol{\pi}^\top \mathbf{1} = 1, \qquad (7)$$

where we normalised the sum of vector entries to 1. Physically, $\boldsymbol{\pi}_j$ is the stationary probability that a surfer is at node $j$. $\boldsymbol{\pi}$ is very expensive to compute for large graphs and the form of $\widetilde{\mathbf{P}}$ invites MC estimation with random walkers. Fogaras et al. (2005) suggest the following algorithm.

**Algorithm 4.1** (Random walks for PageRank estimation). *(Fogaras et al., 2005) Simulate $m \in \mathbb{N}$ runs of a simple random walk with transition probability matrix $\mathbf{P}$ out of every node $i \in \mathcal{N}$, terminating with probability $p$ at each timestep. Evaluate the estimator $\widehat{\boldsymbol{\pi}}_j$ as the fraction of walks terminating at node $j$, $\widehat{\boldsymbol{\pi}}_j := \frac{1}{Nm} \sum_{\mathcal{N}} \sum_{j=1}^{m} \mathbb{I}(\text{walker terminates at } j)$.*

It is straightforward to show that $\widehat{\boldsymbol{\pi}}$ is an unbiased estimator of $\boldsymbol{\pi}$ (see App. A.2). This is a natural setting to test an ensemble of repelling random walks. We are able to make the following surprisingly strong statement.

**Theorem 4.2** (Superiority of repelling random walks for PageRank estimation). *For a transient repelling ensemble,*

$$Var(\widehat{\boldsymbol{\pi}}_j)_{repelling} \leq Var(\widehat{\boldsymbol{\pi}}_j)_{i.i.d.} \tag{8}$$

*for any graph.*

We defer a full proof to App. A.2 but provide a brief sketch below.

**Proof sketch**: Supposing that the number of walkers is smaller than the minimum node degree, the behaviours of a transient repelling and i.i.d. ensemble only differ at the first timestep. In the former scheme walkers are forced to diverge whereas in the latter they are independent. The expectation values of the estimators associated with each walker are conditionally independent given their node positions at $t = 1$ and are identical in both schemes by definition; denote it by $f(v_{t=1})$. With the i.i.d. ensemble the variance depends on $\mathbb{E}_{v^{(1)} \perp v^{(2)}}[f(v_{t=1}^{(1)}) f(v_{t=1}^{(2)})]$ where the node positions of a pair of walkers $v^{(1,2)}$ are independent. Meanwhile, for repelling walkers it depends on $\mathbb{E}_{v^{(1)} \neq v^{(2)}}[f(v_{t=1}^{(1)}) f(v_{t=1}^{(2)})]$ where we condition that $v^{(1,2)}$ cannot be equal. Simple algebra reveals that the latter is smaller. It is straightforward to then generalise to when the number of walkers exceeds the minimum node degree.

It is remarkable that Theorem 4.2 holds for arbitrary $\mathcal{G}$. Table 2 shows the PageRank estimator error with 2 walkers that are either i) i.i.d. or ii) repelling out of every node. The quality of approximation is already excellent with just a single pair. The termination probability is $p = 0.3$ and we take 1000 trials to compute the standard deviations (10000 for `eurosis` since it is larger). As per the theoretical guarantees, repelling random walks consistently perform better.

Table 2: Mean $L_2$-norm of the difference between the true and approximated PageRank vectors $\pi_{\text{err}} := \|\boldsymbol{\pi} - \widehat{\boldsymbol{\pi}}\|_2$, using i.i.d. and repelling pairs of random walkers. Lower is better. Repelling random walks consistently outperform i.i.d. random walks. Parentheses give one standard deviation on the mean error.

| Graph | $N$ | PageRank error, $\pi_{\text{err}}$ | |
| --- | --- | --- | --- |
| | | i.i.d. | repelling |
| Small ER | 20 | 0.0208(2) | **0.0196(2)** |
| Larger ER | 100 | 0.00420(2) | **0.00406(2)** |
| Binary tree | 127 | 0.00290(1) | **0.00270(1)** |
| $d$-regular | 100 | 0.00434(2) | **0.00422(2)** |
| karate | 34 | 0.0124(1) | **0.0115(1)** |
| dolphins | 62 | 0.00686(4) | **0.00651(4)** |
| football | 115 | 0.00385(2) | **0.00376(2)** |
| eurosis | 1272 | 0.000342(2) | **0.000335(2)** |

In the PageRank setting, RRWs are closely related to the algorithm presented by Luo (2019), which was introduced to reduce edge bandwidth. The scheme takes $d_i$ walks out of every node $i$ and permutes them randomly among the neighbours at every timestep. Note that sampling without replacement is identical to permutation if the number of walkers is equal to the number of neighbours to which they must be assigned.

As a brief addendum for the interested reader: $\widehat{\boldsymbol{\pi}}_j$ is actually a member of a broader class of functions coined *step-by-step linear*, defined as follows.

**Definition 4.3** (Step-by-step linear functions). *Let $\Omega_r$ denote the set of all infinite-length walks starting at node $r$, $\Omega_r := \{(v_i)_{i=0}^{\infty} \mid v_0 = r, v_i \in \mathcal{N}, (v_i, v_{i+1}) \in \mathcal{E}\}$. We refer to a function $y : \Omega_r \to \mathbb{R}$ as* step-by-step linear *if it takes the form:*

$$y(\omega) = \sum_{i=0}^{\infty} f(v_i, i) \prod_{j=1}^{i} g(v_{j-1}, v_j, j, j-1), \tag{9}$$

*where $f : \mathcal{N} \times (\mathbb{N} \cup \{0\}) \to \mathbb{R}$ and $g : \mathcal{N} \times \mathcal{N} \times (\mathbb{N} \cup \{0\}) \times (\mathbb{N} \cup \{0\}) \to \mathbb{R}$.*

These functions have the property that the variance of the corresponding Monte Carlo estimator is guaranteed to be suppressed by conditioning that the ensemble of random walks $\{\omega\}_{i=1}^{m}$ is transient repelling. Concretely, the following is true.

**Theorem 4.4** (Variance of step-by-step linear functions is reduced by transient repulsion). *Consider the estimator $Y := \sum_{i=1}^{m} y(\omega_i)$ where $\{\omega_i\}_{i=1}^{m}$ enumerates $m$ (infinite) walks on $\mathcal{G}$ and $y : \Omega_r \to \mathbb{R}$ is a step-by-step linear function. Suppose that the sets of walks $\{\omega_i\}_{i=1}^{m}$ are either i) i.i.d. or ii) transient repelling (Def. 2.2). We have that:*

$$Var(Y_{repelling}) \leq Var(Y_{i.i.d.}). \tag{10}$$

We provide a proof and further discussion in Sec. A.3. Interestingly, the step-by-step linear family also includes $\phi(i)_k$, the $k$th component of the GRF corresponding to the $i$th node of $\mathcal{G}$, though of course this alone is insufficient to guarantee suppression of variance of the kernel estimator $\phi(i)^{\top} \phi(j)$.

## 5 APPLICATION 3: APPROXIMATING GRAPHLET CONCENTRATIONS

Finally, we use repelling random walks to estimate the relative frequencies of *graphlets*: induced subgraph patterns within a graph $\mathcal{G}$. Formally, a $k$-node induced subgraph $\mathcal{G}_k = (\mathcal{V}_k, \mathcal{E}_k)$ satisfies $\mathcal{V}_k \subset \mathcal{V}$, $|\mathcal{V}_k| = k$ and $\mathcal{E}_k = \{(u, v) : u, v \in \mathcal{V}_k \wedge (u, v) \in \mathcal{E}\}$: that is, a subset of $k$ connnected nodes together with any edges between them. For example, for $k = 3$ the possible graphlets are a triangle and a wedge (see Fig. 3). Computing a graph's *graphlet concentrations* – the proportions of different $k$-node graphlets – is

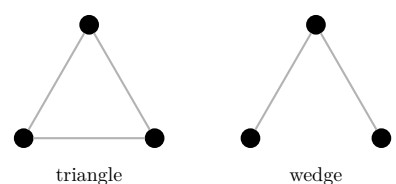

triangle          wedge

Figure 3: Graphlets for $k = 3$

a task of broad interest in biology (Pržulj, 2007; Milenković and Pržulj, 2008) and network science (Becchetti et al., 2008; Ugander et al., 2013) since it characterises the local structure of $\mathcal{G}$ (Milo et al., 2002). Such concentrations even permit construction of *graphlet kernels* $K : \mathcal{G} \times \mathcal{G} \to \mathbb{R}$ to compare different graphs (Shervashidze et al., 2009).

For large graphs, exact computation by exhaustive counting is unfeasible because of the combinatorial explosion in the number of graphlets with $N$. This motivates random walk Markov Chain Monte Carlo approaches. Such crawling-based algorithms also benefit from not requiring access to the entire graph simultaneously: a typical restriction for online social networks where the graph is only available via API calls to retrieve a particular node's neighbours (e.g. user's friends). These algorithms are also easily distributed across machines.

Chen et al. (2016) propose a general algorithm for asymptotically unbiased, efficient estimation of graphlet concentrations using random walks. We summarise one particular instantiation of it for $k = 3$ below.

**Algorithm 5.1** (Graphlet concentration estimation using random walks). *(Chen et al., 2016) Simulate a simple random walk of length $L \in \mathbb{N}$ (the sampling budget) out of a randomly selected node. Consider $X_i^{(3)} = (X_i, X_{i+1}, X_{i+2})$ with $1 \leq i \leq L - 2$, the states of an augmented Markov Chain whose state space is the ordered 3-tuples of consecutively-visited nodes. Discard all such states where $X_i = X_{i+2}$ (where the walker backtracks), and for the*

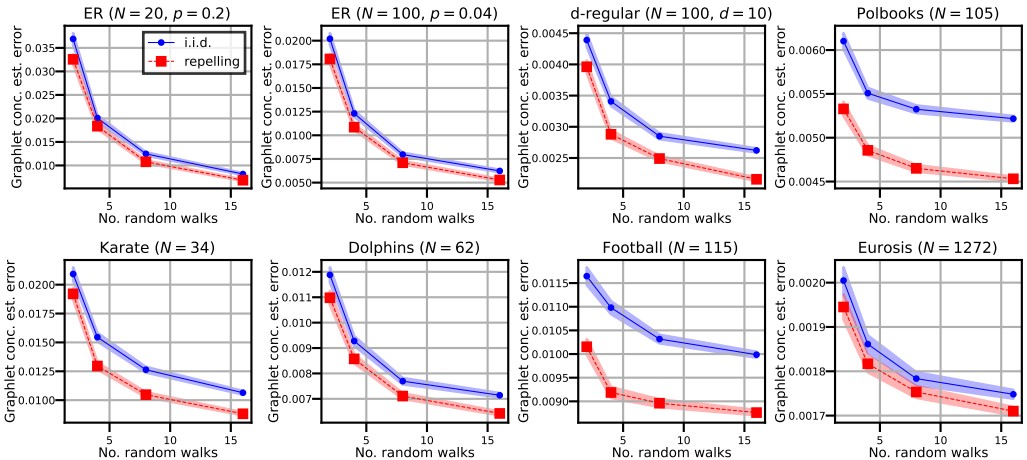

Figure 4: Mean square error on estimates of $k = 3$ graphlet concentrations with different numbers of random walks on different graphs. Lower is better. Using the repelling scheme consistently improves the quality of the estimate compared to independent walks.

*remaining $n$ classify the graphlets $g_i^{(3)}$ to get the weighted counts $C_{wed} \coloneqq \sum_{i=1}^{n-2} \mathbb{I}(g_i^{(3)} = wedge)\frac{d_{i+1}}{2}$ and $C_{tri} \coloneqq \sum_{i=1}^{n-2} \mathbb{I}(g_i^{(3)} = triangle)\frac{d_{i+1}}{6}$ (where $d_{i+1}$ is the degree of the $i + 1$th node). In the limit of large $L$, $\widehat{c}_{tri}^{(3)} \coloneqq \frac{C_{tri}}{C_{tri}+C_{wed}}$ gives an unbiased estimator of the concentration of triangle graphlets.*

The weightings in the computation of $C_{\{wed,tri\}}$ are included to correct for two sources of bias: $d_{i+1}$ accounts for the fact that the stationary distribution of the expanded Markov chain is inversely proportional to the degree of the middle node, $\boldsymbol{\pi}(X_i^{(3)}) = (2|\mathcal{V}|d_{i+1})^{-1}$, and the combinatorial factors adjust for the fact that 6 states $X_i^{(3)}$ correspond to the triangle graphlet (twice the number of Hamiltonian paths) but only 2 correspond to the wedge.

We implement Alg. 5.1 with both i) i.i.d. walkers and ii) a repelling ensemble. A rigorous theoretical analysis of concentration properties is very challenging and is deferred as important future work; for now, our study is empirical. Fig. 4 plots the fractional error of the estimator of triangle graphlet concentration $\widehat{c}_{tri}^{(3)}$ against the number of walkers. We use the same graphs as in Sec. 3, but replace the binary tree with `polbooks` since for the former $\widehat{c}_{tri} = 0$ trivially. We impose a restricted sampling budget with walks of length $L = 16$ to highlight the benefits of repelling random walks in the transient regime, and take 2500 repeats over all starting nodes for statistics. Repelling random walks consistently perform better, providing more accurate estimates of the triangle graphlet concentration, and for some graphs the improvement is large. Alg. 5.1 can be generalised to estimate the concentrations larger graphlets with $k > 3$; we anticipate that repelling random walks will still prove effective.

## 6 Conclusion

We have presented a new quasi-Monte Carlo scheme called *repelling random walks* that induces correlations between the directions of random walkers on a graph. Estimators constructed using this interacting ensemble are guaranteed to remain unbiased but their concentration properties are often substantially improved. We test our algorithm on applications as diverse as estimating graph kernels, the PageRank vector and graphlet concentrations. In every case the experimental performance is very strong and often we are able to provide concrete theoretical guarantees. We hope this work will motivate further research on developing quasi-Monte Carlo methods to improve sampling on graphs.

## 7 Ethics and reproducibility

**Ethics**: Our work is foundational with no immediate ethical concerns apparent to us. However, increases in scalability provided by quasi-Monte Carlo algorithms could exacerbate existing and incipient risks of graph-based machine learning, from bad actors or as unintended consequences.

**Reproducibility**: Every effort has been made to guarantee the work's reproducibility. The core algorithm is clearly presented in Def. 2.1. Accompanying theoretical results are proved and discussed in the Appendices A.1-A.3, including any assumptions where appropriate. Source code is available at https://github.com/isaac-reid/repelling_random_walks. All datasets used correspond to standard graphs and are freely available online; we give links to suitable repositories in every instance. Results are reported with uncertainties to facilitate comparison.

## 8 Relative contributions and acknowledgements

IR conceived and implemented the repelling random walks mechanism, derived the theoretical results and prepared the manuscript. KC was crucially involved in the project throughout, acting as the senior research lead. EB proposed the class of step-by-step linear functions and made important theoretical contributions. AW provided helpful discussions and guidance.

IR acknowledges support from a Trinity College External Studentship. AW acknowledges support from a Turing AI fellowship under grant EP/V025279/1 and the Leverhulme Trust via CFI.

We thank Kenza Tazi and Austin Tripp for their careful readings of the manuscript, and Yashar Ahmadian for interesting discussions about the relationship with quantum entanglement. Richard Turner provided valuable suggestions and support throughout the project.

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

# A  APPENDICES

## A.1  PROOF OF THEOREM 3.1 (SUPERIORITY OF REPELLING RANDOM WALKS FOR KERNEL ESTIMATION

In this appendix, we prove Theorem 3.1: namely, that using transient repelling random walks reduces the mean square error of graph random feature (GRF) estimates of the 2-regularised Laplacian kernel, defined by:

$$\left[\widehat{\mathbf{K}}_{\text{lap}}^{(2)}\right]_{ij} := \left(\mathbf{I} - \widetilde{\mathbf{L}}\right)_{ij}^{-2} = (1 + \sigma^2)^{-2}\left(\mathbf{I} - \frac{\sigma^2}{1 + \sigma^2}\mathbf{W}\right)^{-2} \tag{11}$$

where $\widetilde{\mathbf{L}}$ is the Laplacian, $\sigma$ is a regulariser and $\mathbf{W}$ is a normalised weighted adjacency matrix with elements $\mathbf{W} = \left[a_{ij}/(\tilde{d}_i\tilde{d}_j)^{1/2}\right]_{i,j=1}^N$ (with $\tilde{d}_i = \sum_j a_{ij}$ the degree of the $i$th node and $a_{ij}$ the weight of the edge before normalisation). Ignoring the overall normalisation constant and absorbing the factor of $\sigma^2/(1 + \sigma^2)$ into $\mathbf{W}$, wlg we will now consider estimation of

$$\widehat{\mathbf{K}}_{ij} = (\mathbf{I} - \mathbf{W})^{-2} \tag{12}$$

where $\mathbf{W} = [w_{ij}]_{i,j=1}^N$.

Directly from the definition of the GRF vector (see e.g. (Choromanski, 2023) or (Reid et al., 2023a)), we have that

$$\widehat{\mathbf{K}}_{ij} = \phi(i)^\top\phi(j) = \frac{1}{m^2}\sum_{x\in\mathcal{N}}\sum_{\omega_{ix}\in\Omega_{ix}}\sum_{\omega_{jx}\in\Omega_{jx}}\frac{\widetilde{\omega}(\omega_{ix})}{p(\omega_{ix})}\frac{\widetilde{\omega}(\omega_{jx})}{p(\omega_{jx})}N(\omega_{ix})N(\omega_{jx}) \tag{13}$$

where

$$N(\omega_{ix}) := \sum_{l=1}^m\mathbb{I}\left(\omega_{ix}\in\bar{\Omega}_l\right). \tag{14}$$

Here: $m\in\mathbb{N}$ is the number of random walkers simulated out of each node; $\Omega_{ix}$ is the set of all walks on the graph between the nodes indexed $i$ and $x$; $\omega_{ix}$ is a member of this set; $\widetilde{\omega}(\omega)$ is a function that returns the product of weights of edges traversed by a graph random walk $\omega$; and $p(\omega)$ is a function that returns the marginal probability of a random walk (equal to $((1-p)/d)^{\text{len}(\omega)}$ – with $0 < p < 1$ a finite termination probability, $d$ the node degree and $\text{len}(\omega)$ the walk length – in the case of a $d$-regular graph). $\bar{\Omega}_l$ denotes the $l$th walk out of node $i$ such that $N(\omega_{ix})$ counts the empirical number of walkers completing a particular prefix subwalk $\omega_{ix}$, a discrete random variable between 0 and $m$. Since $\mathbb{E}(N(\omega_{ix})) = mp(\omega_{ix})$, it is straightforward to see that

$$\mathbb{E}(\phi(i)^\top\phi(j)) = \sum_{x\in\mathcal{N}}\sum_{\omega_{ix}\in\Omega_{ix}}\sum_{\omega_{jx}\in\Omega_{jx}}\widetilde{\omega}(\omega_{ix})\widetilde{\omega}(\omega_{jx})$$
$$= \sum_{\omega_{ij}\in\Omega_{ij}}(\text{len}(\omega_{ij}) + 1)\widetilde{\omega}(\omega_{ij}) = (\mathbf{I} - \mathbf{W})^{-2} \tag{15}$$

which confirms that the estimator is unbiased. Our task is now to determine how the variance of $\widehat{\mathbf{K}}_{ij}$ depends on whether the ensemble of $m$ walkers from each node is i) independent or ii) repelling. We will see that, under some conditions, it is guaranteed to be smaller in the latter case.

The following is true:

$$\mathbb{E}\left(\widehat{\mathbf{K}}_{ij}^2\right) = \frac{1}{m^4}\sum_{x,y\in\mathcal{N}}\left[\sum_{\omega_{ix},\omega_{iy}}\frac{\widetilde{\omega}(\omega_{ix})}{p(\omega_{ix})}\frac{\widetilde{\omega}(\omega_{iy})}{p(\omega_{iy})}\mathbb{E}(N(\omega_{ix})N(\omega_{iy}))\right]$$
$$\cdot\left[\sum_{\omega_{jx},\omega_{jy}}\frac{\widetilde{\omega}(\omega_{jx})}{p(\omega_{jx})}\frac{\widetilde{\omega}(\omega_{jy})}{p(\omega_{jy})}\mathbb{E}(N(\omega_{jx})N(\omega_{jy}))\right]. \tag{16}$$

This makes clear that the object of central importance will be

$$\mathbb{E}\left(N(\omega_{ix})N(\omega_{iy})\right) \tag{17}$$

where $x, y \in \mathcal{N}$. We will now consider how this depends on i) the pair of subwalks $(\omega_{ix}, \omega_{iy})$ and ii) the presence or absence of repulsion.

To avoid notational clutter, we will write expressions as if the graph is $d$-regular with the understanding that it is trivial to relax this without changing any conclusions, making the replacement: $d^{\mathrm{len}(\omega_{ix})} \to \prod_{i=0}^{\mathrm{len}(\omega_{ix})-1} d_i$.

**i.i.d. walkers:** Begin with the simpler i.i.d. case. First consider the case that $\omega_{ix} \neq \omega_{iy}$, $\omega_{ix} \notin \omega_{iy}$ and $\omega_{iy} \notin \omega_{ix}$: namely, that the walks are distinct and neither is a (strict) subwalk of the other. It follows that a single walker cannot take both subwalks simultaneously. We also assume that both walks are of length $\mathrm{len}(\omega_{i(x,y)}) \geq 1$. Then we have that

$$\mathbb{E}\left[N(\omega_{ix})N(\omega_{iy})\right] = \mathbb{E}\left[\sum_{l_1=1}^{m}\sum_{l_2=1}^{m}\mathbb{I}(\omega_{ix}\in\bar{\Omega}_{l_1})\mathbb{I}(\omega_{iy}\in\bar{\Omega}_{l_2}))\right]$$
$$= m(m-1)\left(\frac{1-p}{d}\right)^{\mathrm{len}(\omega_{ix})+\mathrm{len}(\omega_{iy})}. \tag{18}$$

What about if $\omega_{ix} \in \omega_{iy}$? It is straightforward to convince oneself that

$$\mathbb{E}\left[N(\omega_{ix})N(\omega_{iy})\right] = m(m-1)\left(\frac{1-p}{d}\right)^{\mathrm{len}(\omega_{ix})+\mathrm{len}(\omega_{iy})} + m\left(\frac{1-p}{d}\right)^{\mathrm{len}(\omega_{iy})} \tag{19}$$

where the extra second correlation term comes from a single walker completing both subwalks. Lastly, suppose that $\mathrm{len}(\omega_{ix}) = 0$ (i.e. one of the subwalks has zero length). Then we have that

$$\mathbb{E}\left[N(\omega_{ix})N(\omega_{iy})\right] = m^2\left(\frac{1-p}{d}\right)^{\mathrm{len}(\omega_{iy})}. \tag{20}$$

Now we move onto the repelling case, which is substantially more difficult.

**Repelling walkers:** For tractability, we will consider the transient repulsion scheme described in Def. 2.2. Suppose that we have $N'_\alpha$ walkers at some node indexed $\alpha \neq i$, with the set of neighbouring nodes including nodes labelled $\beta$ and $\gamma$ (i.e. $\beta, \gamma \in \mathcal{N}(\alpha)$). An important quantity is

$$\mathbb{E}(N_\beta N_\gamma | N'_\alpha). \tag{21}$$

From Def. 2.1, we have that

$$N_\beta = N_\alpha // d + \epsilon_1, \qquad N_\gamma = N_\alpha // d + \epsilon_2, \tag{22}$$

where $//$ denotes truncating integer division $\epsilon_{1,2}$ are anticorrelated binary random variables. The reason they are anticorrelated is that a walker that transitions to $\beta$ cannot also transition to $\gamma$. With a little work, one can convince oneself that

$$\mathbb{E}(N_\beta N_\gamma | N'_\alpha) = \left(\frac{N'_\alpha}{d}\right)^2 + \frac{R}{d}\left(\frac{R-1}{d-1} - \frac{R}{d}\right), \tag{23}$$

where $R := N'_\alpha \% d$, the remainder after the the walkers have been partitioned into blocks of size $d$. From this form, we can see that we will be concerned with the statistics of $N'_\alpha$: in particular how $\mathbb{E}(N'^2_\alpha)$ behaves. Understanding this is our next task.

Let $N_\alpha$ be a random variable denoting the number of walkers at node $\alpha$ of some particular walk on the graph. Then let $N'_\alpha$ denote the number of walkers surviving the '$p$-step', where each walker terminates independently with probability $p$. Let $N_\beta$ denote the number of walkers that subsequently hop to node $\beta$ on the walk. It is clear that $N_\beta = N // d + \epsilon$, with $\epsilon$

a random variable that takes a value of 1 with probability $\frac{R}{d}$ and 0 with probability $1 - \frac{R}{d}$. It is simple to show that

$$\mathbb{E}(N_\beta^2) = \frac{\mathbb{E}(N_\alpha'^2)}{d^2} + \mathbb{E}\left(\frac{R}{d}(1 - \frac{R}{d})\right). \tag{24}$$

It is also trivially the case that $\mathbb{E}(N_\alpha'^2) = \mathbb{E}(N_\alpha^2)(1-p)^2 + \mathbb{E}(N_\alpha)p(1-p)$ (law of iterated expectations). The second term in Eq. 24 is generically difficult to describe analytically, but it is simple in the special case that $N_\alpha' < d$ so $R = N_\alpha'$. In particular, here we have that

$$\mathbb{E}(N_\alpha^2) = \mathbb{E}(N_\alpha) = m\left(\frac{1-p}{d}\right)^{\text{len}(\omega_{i\alpha})} \tag{25}$$

whereupon, referring back to Eq. 23,

$$\mathbb{E}(N_\beta N_\gamma | N_\alpha') = 0. \tag{26}$$

It is trivial to see why this must be the case: supposing we begin with fewer than $d$ walkers at node $i$, in the transient repelling scheme they all diverge at the first timestep. Any subsequent node $\alpha$ on some walk is occupied by at most 1 walker which chooses one of its $d_\alpha$ neighbours at random, so value of $N_\beta N_\gamma$ (product of occupations of its child nodes) always vanishes. It is encouraging that our algebraic approach reproduces this intuitive result. It follows immediately that, for walks diverging at some node *not* equal to the starting node $i$, $\mathbb{E}\left[N(\omega_{ix})N(\omega_{iy})\right] = 0$.

What about if the walkers instead diverge at $i$? Here the result is different because $\mathbb{E}(N_i) = m$, the initial number of particles, and $\text{Var}(m) = 0$ since the total number of walkers is a fixed hyperparameter. After just a little work,

$$\mathbb{E}(N_\beta N\gamma) = \frac{(1-p)^2}{d(d-1)}m(m-1) \tag{27}$$

and therefore

$$\mathbb{E}\left[N(\omega_{ix})N(\omega_{iy})\right] = \frac{d}{d-1}m(m-1)\left(\frac{1-p}{d}\right)^{\text{len}(\omega_{ix})+\text{len}(\omega_{iy})}. \tag{28}$$

This is also intuitive: the repelling scheme shifts probability mass onto walks that diverge at $i$, enhancing this correlation term.

The subwalk case is also straightforward. Supposing $\omega_{ix} \in \omega_{iy}$, we can use Eq. 25 to show that

$$\mathbb{E}\left[N(\omega_{ix})N(\omega_{iy})\right] = m\left(\frac{1-p}{d}\right)^{\text{len}(\omega_{iy})} \tag{29}$$

because if the walkers immediately diverge then we can only sample both $\omega_{iy}$ and $\omega_{ix} \in \omega_{iy}$ if a single walk traverses both. Lastly, if $\text{len}(\omega_{ix}) = 0$, we still have that

$$\mathbb{E}\left[N(\omega_{ix})N(\omega_{iy})\right] = m^2\left(\frac{1-p}{d}\right)^{\text{len}(\omega_{iy})} \tag{30}$$

which is natural because if one (or both) of the walks is of zero length then the repulsion scheme cannot modify the correlation term.

We summarise these observations in Table 3, denoting $c := \frac{1-p}{d}$ and $\omega_{ix}$ as shorthand for $\text{len}(\omega_{ix})$ for compactness.

Table 3

| Class | $\mathbb{E}(N(\omega_{ix})N(\omega_{iy}))$ | |
| --- | --- | --- |
| | i.i.d. | transient repelling |
| Same walk, $\omega_{ix} = \omega_{iy}$ | $mc^{\omega_{ix}} + m(m-1)c^{2\omega_{ix}}$ | $mc^{\omega_{ix}}$ |
| Subwalk, $\omega_{ix} \in \omega_{iy}$ | $mc^{\omega_{iy}} + m(m-1)c^{\omega_{ix}+\omega_{iy}}$ | $mc^{\omega_{iy}}$ |
| Different walks, diverge at $i$ | $m(m-1)c^{\omega_{ix}+\omega_{iy}}$ | $\frac{d}{d-1}m(m-1)c^{\omega_{ix}+\omega_{iy}}$ |
| Different walks, diverge at $d \neq i$ | $m(m-1)c^{\omega_{ix}+\omega_{iy}}$ | $0$ |
| $\text{len}(\omega_{ix}) = 0$ | $m^2 c^{\omega_{iy}}$ | $m^2 c^{\omega_{iy}}$ |

More explicitly, we can write

$$
\mathbb{E}(N(\omega_{ix})N(\omega_{iy})) =
\begin{cases}
\begin{aligned}
& m(m-1)c^{\omega_{ix}+\omega_{iy}}\mathbb{I}(\text{both} > 0) \\
& +mc^{\omega_{\text{longer}}}\mathbb{I}(\text{subwalks, both} > 0) \\
& \qquad +m^2 c^{\omega_{\text{longer}}}\mathbb{I}(\text{len } 0)
\end{aligned} & \text{if i.i.d.} \\[4ex]
\begin{aligned}
& \frac{d_i}{d_i-1}m(m-1)c^{\omega_{ix}+\omega_{iy}}\mathbb{I}(\text{both} > 0, \text{div at } i) \\
& \qquad +mc^{\omega_{\text{longer}}}\mathbb{I}(\text{subwalks, both} > 0) \\
& \qquad +m^2 c^{\omega_{\text{longer}}}\mathbb{I}(\text{len } 0)
\end{aligned} & \text{if repelling}
\end{cases}
\tag{31}
$$

where *only the first term differs*. Here, $\mathbb{I}(\text{both} > 0)$ means both walks traverse at least one edge, $\mathbb{I}(\text{len } 0)$ means one of the walks is of length 0, $\mathbb{I}(\text{subwalks})$ means $\omega_{ix} \in \omega_{iy}$ (or vice versa), and $\omega_{\text{longer}}$ denotes the length of the longer walk. We will now insert these expressions into Eq. 16 to compute the difference in variance of $\widehat{\mathbf{K}}_{ij}$ with the two possible coupling schemes.

After tedious but straightforward algebra, we arrive at the following closed form:

$$
\mathrm{Var}\left(\left[\widehat{\mathbf{K}}_{ij}\right]_{\text{i.i.d.}}\right) - \mathrm{Var}\left(\left[\widehat{\mathbf{K}}_{ij}\right]_{\text{repelling}}\right) =
$$

$$
\left.
\begin{aligned}
& \frac{(m-1)^2}{m^2}\left(\left[\frac{\mathbf{W}^2}{(1-\mathbf{W})^2}\right]_{ij}\right)^2 \\
-\frac{d_i d_j}{(d_i-1)(d_j-1)}\frac{(m-1)^2}{m^2}\sum_{\substack{i' \in \mathcal{N}(i) \\ i'' \in \mathcal{N}(i)\setminus i'}}\sum_{\substack{j' \in \mathcal{N}(j) \\ j'' \in \mathcal{N}(j)\setminus j'}} & w_{ii'}w_{jj'}w_{ii''}w_{jj''}\left[\frac{1}{(1-\mathbf{W})^2}\right]_{i'j'}\left[\frac{1}{(1-\mathbf{W})^2}\right]_{i''j''}
\end{aligned}
\right\} \text{(a)}
$$

$$
\left.
\begin{aligned}
& +\frac{d_i}{d_i-1}\frac{m-1}{m^2}\sum_{x}\sum_{\omega_{ix}>0}\frac{\widetilde{\omega}(\omega_{ix})^2}{p(\omega_{ix})}\left[B(x,j)-C(x,j)\right] \\
& +\frac{d_j}{d_j-1}\frac{m-1}{m^2}\sum_{x}\sum_{\omega_{jx}>0}\frac{\widetilde{\omega}(\omega_{jx})^2}{p(\omega_{jx})}\left[B(x,i)-C(x,i)\right] \\
& +\frac{m-1}{m}\left[\frac{d_i}{d_i-1}\left(B(i,j)-C(i,j)\right)+\frac{d_j}{d_j-1}\left(B(j,i)-C(j,i)\right)\right]
\end{aligned}
\right\} \text{(b)}
$$

$$
\tag{32}
$$

where

$$
B(x,i) := \sum_{i' \in \mathcal{N}(i)} w_{ii'}^2\left[\frac{1}{(1-\mathbf{W})^2}\right]_{xi'}^2 - \frac{1}{d_i}\left(\sum_{i' \in \mathcal{N}(i)}w_{ii'}\left[\frac{1}{(1-\mathbf{W})^2}\right]_{xi'}\right)^2
\tag{33}
$$

and

$$
C(x,i) := \sum_{i' \in \mathcal{N}(i)} w_{ii'}^2\left[\frac{\mathbf{W}}{(1-\mathbf{W})^2}\right]_{xi'}^2 - \frac{1}{d_i}\left(\sum_{i' \in \mathcal{N}(i)}w_{ii'}\left[\frac{\mathbf{W}}{(1-\mathbf{W})^2}\right]_{xi'}\right)^2.
\tag{34}
$$

Note that both $B$ and $C$ are always positive by Jensen's inequality. It is remarkable that such a simple expression exists for the difference in kernel estimator variance between the i.i.d. and transient repelling schemes.

Showing the class of graphs for which this is guaranteed to be positive is a challenging open problem, but we are able to make progress in some tractable special cases.

For instance, consider the limit $w \to 0$ with all the graph weights equal, $\mathbf{W} = w\mathbf{A}$. In this case, when computing matrix elements we can just retain terms corresponding to the shortest path. For example,

$$
\left[\frac{1}{(1-w\mathbf{A})^2}\right]_{ij} = \left[1 + 2w\mathbf{A} + 3w^2\mathbf{A}^2 + ...\right]_{ij} = M(l_{ij})(l_{ij}+1)w^{l_{ij}} + \mathcal{O}(w^{l_{ij}+1})
\tag{35}
$$

where $l_{ij}$ denotes the length of the shortest path between nodes $i$ and $j$ and $M(l_{ij})$ denotes its *multiplicity*: the number of such unique paths that exist in $\mathcal{G}$. To give examples, for a tree $M(l_{ij}) = 1$ and for a square grid $M(l_{ij}) = \binom{a+b}{a}$, where $a$ is the difference in $x$-coordinates of nodes $i$ and $j$ and $b$ is the difference in $y$ coordinates.

**When will (a) be positive?** Provided that nodes $i$ and $j$ are separated by at least 2 edges, the following is true:

$$\frac{(m-1)^2}{m^2} \left( \left[ \frac{w^2 \mathbf{A}^2}{(1-w\mathbf{A})^2} \right]_{ij} \right)^2 = \frac{(m-1)^2}{m^2} M(l_{ij})^2 (l_{ij}-1)^2 w^{2l_{ij}} + \mathcal{O}(w^{2l_{ij}+1}), \qquad (36)$$

and

$$\frac{d_i d_j}{(d_i - 1)(d_j - 1)} \frac{(m-1)^2}{m^2} w^4 \sum_{\substack{i' \in \mathcal{N}(i) \\ i'' \in \mathcal{N}(i) \setminus i'}} \sum_{\substack{j' \in \mathcal{N}(j) \\ j'' \mathcal{N}(j) \setminus j'}} \left[ \frac{1}{(1-w\mathbf{A})^2} \right]_{i'j'} \left[ \frac{1}{(1-w\mathbf{A})^2} \right]_{i''j''}$$

$$= \frac{d_i d_j}{(d_i - 1)(d_j - 1)} \frac{(m-1)^2}{m^2} \sum_{\substack{i' \in \mathcal{N}(i) \\ i'' \in \mathcal{N}(i) \setminus i'}} \sum_{\substack{j' \in \mathcal{N}(j) \\ j'' \mathcal{N}(j) \setminus j'}} M(l_{i'j'}) M(l_{i''j''}) w^{l_{i'j'} + l_{i''j''} + 4} (l_{i'j'} + 1)(l_{i''j''} + 1)$$

$$+ \mathcal{O}(w^{l_{i'j'} + l_{i''j''} + 5}).$$

$$(37)$$

Note that, since $i' \in \mathcal{N}(i)$ and $j' \in \mathcal{N}(j)$, it is trivially the case that $l_{ij} - 2 \leq l_{i'j'} \leq l_{ij} + 2$. In Eq. 37, only terms where $l_{i'j'} = l_{i''j''} = l_{ij} - 2$ will give contributions of the same order as the leading term $\mathcal{O}(w^{2l_{ij}})$ in Eq. 36. The condition that Eq. 36 is greater at the leading order is then:

$$M(l_{ij})^2 \geq \frac{d_i d_j}{(d_i - 1)(d_j - 1)} \sum_{\substack{i' \in \mathcal{N}(i), i'' \in \mathcal{N}(i) \setminus i', j' \in \mathcal{N}(j), j'' \in \mathcal{N}(j) \setminus j' \\ l_{i'j'} = l_{i''j''} = l_{ij} - 2}} M(l_{i'j'}) M(l_{i''j''}) \qquad (38)$$

where we remind the reader that $M(l_{ij})$ denotes the degeneracy (number) of shortest paths (of length $l_{ij}$) between nodes $i$ and $j$. This encodes the topological constraint that is sufficient for variance reduction with repelling random walkers on an equal-weights graph as $w \to 0$. Heuristically, the set of nodes $\mathcal{N}(i)$ cannot be too connected to the nodes $\mathcal{N}(j)$.

A cursory numerical check suggests that Eq. 38 is not generically satisfied for every pair of nodes $(i, j)$ on arbitrary graphs, but it does seem to very often be true. We can, however, identify some particular examples where it is guaranteed to hold. For example, it is trivially true for trees for which $M(l_{ij}) = 1$ but the RHS is 0. To wit: for trees there is a unique shortest path between nodes $i$ and $j$ and it is only the case that $l_{i'j'} = l_{ij} - 2$ if both $i'$ and $j'$ lie on this path. Then conditioning that $i'' \neq i'$ and $j'' \neq j'$ means that we cannot fulfil $l_{i''j''} = l_{ij} - 2$, whereupon the sum is over the empty set so evaluates to 0. It is also true for the two dimensional square grid. Without loss of generality, locate node $i$ at coordinates $(0,0)$ and node $j$ at $(a,b)$. If $a = 0$ or $b = 0$ there is a unique shortest path so the inequality follows trivially. If $a = 1$ then $M(l_{ij}) = b + 1$ whereas the RHS evaluates to $2 \left( d/(d-1) \right)^2$ which is smaller for $b \geq 1$ and $d = 4$. Finally, if $a, b > 1$, then $M(l_{ij}) = \binom{a+b}{a}$ (a walker on a shortest path between nodes $i$ and $j$ must take $a$ steps in one direction and $b$ steps in the other, but we are free to permute their order). Meanwhile, the sum on the RHS of Eq. 38 evaluates to:

$$\left( \frac{d}{d-1} \right)^2 \cdot 2 \cdot \left( \binom{a+b-2}{a-2} \binom{a+b-2}{b-2} + \binom{a+b-2}{a-1}^2 \right) \qquad (39)$$

whereupon the ratio RHS/LHS is:

$$\left( \frac{d}{d-1} \right)^2 \cdot 2 \cdot \frac{a(a-1)b(b-1) + a^2 b^2}{[(a+b)(a+b-1)]^2}$$

$$= \left( \frac{d}{d-1} \right)^2 \cdot \frac{2ab}{(a+b)^2} \left[ 1 + \frac{a(a-1) + b(b-1)}{(a-1)(b-1) + ab} \right]^{-1} \overset{?}{\leq} 1 \qquad (40)$$

The expression in square brackets is greater than 1 so its inverse is smaller than 1. Hence, it is sufficient that

$$\frac{(a+b)^2}{2ab} = \frac{(a-b)^2}{2ab} + 2 \geq \left(\frac{d}{d-1}\right)^2 \tag{41}$$

which is trivially always true for $d = 4$. The inequality in Eq. 38 then holds in all cases where nodes $i$ and $j$ are separated by at least 2 edges, so terms (a) will indeed be positive on a two-dimensional square grid.

Having asserted that the terms labelled (a) in Eq. 32 will sum to a positive value under certain conditions (namely: graphs characterised by Eq. 38 with equal edge weights $w \to 0$, considering nodes $i$ and $j$ separated by at least 2 edges), we now proceed to consider the terms labelled (b).

**When will (b) be positive?** Now we consider the remaining terms involving e.g. $B(x,i) - C(x,i)$ where $x, i \in \mathcal{N}$. These demand a little more care because the sum over $x \in \mathcal{N}$ means that we need to account for terms where $x = i$ even when considering off-diagonal terms of the kernel estimate $i \neq j$. Note that

$$B(x,i) = d_i w^2 \text{Var}_{i' \in \mathcal{N}(i)} \left[(1 - w\mathbf{A})^{-2}\right]_{xi'}, \tag{42}$$

the empirical variance of the matrix elements $\mathbf{K}_{xi'}$ among the set of vertices $i'$ that neighbour $i$.

Denote by $\tilde{l}_{xi}$ the smallest walk length for which the variance of the number of walks from $x$ to the set of nodes $\mathcal{N}(i)$ is nonzero,

$$\tilde{l}_{xi} = \min_{l \in \mathbb{N}} \left(\{l \mid \text{Var}_{i' \in \mathcal{N}(i)}(\mathbf{A}_{xi'}^l) \neq 0\}\right). \tag{43}$$

This might well correspond to the shortest path between $x$ and $i' \in \mathcal{N}(i)$, but this is not necessarily the case (i.e. if all the neighbours $i'$ have an equal number of equally short paths to $x$ so the variance on this quantity vanishes). Then we have that

$$B(x,i) = d w_i^2 \text{Var}_{i' \in \mathcal{N}(i)} \left[(1 - w\mathbf{A})^{-2}\right]_{xi'} = d(\tilde{l}_{xi} + 1)^2 w^{2\tilde{l}_{xi}} \text{Var}_{i' \in \mathcal{N}(i)} \left[\mathbf{A}_{xi'}^{\tilde{l}_{xi}}\right] + \mathcal{O}(w^{2\tilde{l}_{xi}+1}) \tag{44}$$

and

$$C(x,i) = d_i w^2 \text{Var}_{i' \in \mathcal{N}(i)} \left[w\mathbf{A}(1 - w\mathbf{A})^{-2}\right]_{xi'} = d(\tilde{l}_{xi})^2 w^{2\tilde{l}_{xi}} \text{Var}_{i' \in \mathcal{N}(i)} \left[\mathbf{A}_{xi'}^{\tilde{l}_{xi}}\right] + \mathcal{O}(w^{2\tilde{l}_{xi}+1}) \tag{45}$$

where $\text{Var}_{i' \in \mathcal{N}(i)} \left[\mathbf{A}_{xi'}^{\tilde{l}_{xi}}\right]$ denotes this first nonvanishing variance. For trees with $x \neq i$, $\tilde{l}_{xi} = l_{xi} - 1$ and $\text{Var}_{i' \in \mathcal{N}(i)} \left[\mathbf{A}_{xi'}^{\tilde{l}_{xi}}\right] = \frac{d-1}{d^2}$.

To leading order in $w$, it is trivial to see that $B(x,i) > C(x,i)$. Inserting back into Eq. 32 and noting the positivity of the prefactor $\frac{\tilde{\omega}(\omega_{ix})^2}{p(\omega_{ix})}$, the positivity of (2) follows. Note that in this section we have not assumed anything about the structure of $\mathcal{G}$ beyond equal weights and $w \to 0$. The topological constraints originate solely from the terms in (a).

Combining the above arguments, we conclude that the using the transient repelling scheme is indeed guaranteed to suppress the variance of kernel estimators $\widehat{\mathbf{K}}_{ij}$ for nodes $i$ and $j$ separated by at least 2 edges under certain conditions: namely, that we have an equally-weighted graph with $w \to 0$, and that the topological condition in Eq. 38 is true (which is the case for e.g. trees and two-dimensional grids). $\square$

We stress that, in practice, the scheme performs very well even for much more general classes of graphs and in the non-asymptotic $w$ limit. We defer extending the proof above to include these cases as future work.

## A.2 Proof of Theorem 4.2 (superiority of repelling random walks for PageRank estimation

In this appendix, we show how random walks can be used to estimate to the PageRank vector and prove that using a transient repelling ensemble reduces the estimator mean square error (Theorem 4.2).

Our intention is to estimate the vector $\boldsymbol{\pi}$, defined as the stationary distribution of the transition matrix $\widetilde{\mathbf{P}}$ defined in Eq. 6 and reproduced below:

$$\widetilde{\mathbf{P}} := (1 - p)\mathbf{P} + \frac{p}{N}\mathbf{E}. \tag{46}$$

The reader should refer back to Eq. 6 for all symbol definitions. We then require that

$$\boldsymbol{\pi}^\top \widetilde{\mathbf{P}} = \boldsymbol{\pi}^\top, \quad \boldsymbol{\pi}^\top \mathbf{1} = 1. \tag{47}$$

Rearranging and Taylor expanding $(1 - (1-p)\mathbf{P})^{-1}$, it is straightforward to convince oneself that the solution is given by

$$\boldsymbol{\pi}_i = \frac{p}{N} \sum_{j \in \mathcal{N}} \sum_{k=0}^{\infty} (1-p)^k \mathbf{P}_{ji}^k. \tag{48}$$

This is nothing other than a sum over all walks $\omega_{ji}$ from each of the graph nodes $j$ to node $i$, each weighted by a factor of $\left(\frac{1-p}{d}\right)^k p$ (with $d^k$ generalising to the product of node degrees along $\omega_{ji}$ if the graph is not $d$-regular) and normalised by the number of vertices $N$. Equivalently, supposing we simulate a random walk out of a random node on the graph $j$, it is the probability that it terminates at node $i$. This invites the algorithm proposed by Fogaras et al. (2005) and shown in Alg. 4.1. We construct the unbiased estimator

$$\widehat{\boldsymbol{\pi}}_i = \frac{1}{Nm} \sum_{j \in \mathcal{N}} \sum_{l=1}^{m} \mathbb{I}[\Omega_l^{(j)} \text{ terminates at node } i] \tag{49}$$

where $\Omega_l^{(j)}$ denotes the $l$th walk (out of a total of $m \in \mathbb{N}$) simulated from node $j$.

Our task is now to consider the variance properties of the estimator $\widehat{\boldsymbol{\pi}}$ when ensembles of walkers out of each node are either i) independent or ii) repelling according to our QMC scheme defined in Def. 2.1. Evidently,

$$\mathbb{E}(\widehat{\boldsymbol{\pi}}_i^2) = \frac{1}{N^2 m^2} \sum_{j_1, j_2 \in \mathcal{N}} \sum_{l_1, l_2 = 1}^{m} \mathbb{E}\left\{\mathbb{I}[\Omega_{l_1}^{(j_1)} \text{ terminates at node } i\,] \, \mathbb{I}[\Omega_{l_2}^{(j_2)} \text{ terminates at node } i\,]\right\}. \tag{50}$$

By construction, only walkers out of the same node are correlated: we simulate repelling ensembles out of every vertex but a pair of walkers coming from *different* vertices remain independent throughout. Therefore, it is sufficient to consider the behaviour of terms $j_1 = j_2$. In particular, we we need to determine whether the value of

$$\mathbb{E}\left\{\mathbb{I}[\Omega_{l_1}^{(j)} \text{ terminates at node } i\,] \, \mathbb{I}[\Omega_{l_2}^{(j)} \text{ terminates at node } i\,]\right\} \tag{51}$$

is suppressed with repelling random walks for fixed arbitrary $j \in \mathcal{N}$. We will refer to this as the *correlation term*.

First consider the special case $j \neq i$ so all walks are of length at least 1. We also consider *transient repulsion* (see Def. 2.2) so that walkers only repel at the first timestep; the full-repelling scheme is empirically effective but difficult to reason about analytically.

For i.i.d. walkers, the correlation term in Eq. 51 evaluates to

$$p^2 (1-p)^2 \left[\frac{\mathbf{P}}{1 - (1-p)\mathbf{P}}\right]_{ji}^2 = p^2 \left(\frac{1-p}{d_j}\right)^2 \sum_{j' \in \mathcal{N}(j)} \sum_{j'' \in \mathcal{N}(j)} \left[\frac{1}{1 - (1-p)\mathbf{P}}\right]_{j'i} \left[\frac{1}{1 - (1-p)\mathbf{P}}\right]_{j''i} \tag{52}$$

At the first timestep, a pair of repelling walkers are either assigned to the same 'block' (see Fig. 1) or different 'blocks' with a probability that depends on the number of walkers $m$ and the degree of the first node $d_j$. The correlation term in Eq. 51 depends on its evaluations conditioned on each of these two events, weighted by their respective probabilities.

If the pair are assigned to different blocks their dynamics are i.i.d. so the correlation term is unmodified. Meanwhile, if they are assigned to the same block (which happens almost

surely if $m \leq d_j$), then the equivalent term evaluates to

$$p^2 \left(\frac{1-p}{d_j}\right)^2 \left(\frac{d_j}{d_j - 1}\right) \sum_{j' \in \mathcal{N}(j)} \sum_{j'' \in \mathcal{N}(j) \setminus j'} \left[\frac{1}{1 - (1-p)\mathbf{P}}\right]_{j'i} \left[\frac{1}{1 - (1-p)\mathbf{P}}\right]_{j''i}. \quad (53)$$

This differs from Eq. 52 in that i) the variable $j''$ in the sum over the neighbours of $j$ can no longer be equal to $j'$ since the walks repel, and ii) there is an extra factor of $\frac{d_j}{d_j - 1}$ to account for the increase the conditional probability of choosing $j''$ given that $j'$ becomes excluded when the first walker picks it ('without replacement').

Denote the matrix element $f(j', i) := \left[\frac{1}{1-(1-p)\mathbf{P}}\right]_{j'i}$. Then the difference between the terms in Eqs 52 and 53 is equal to

$$p^2 \left(\frac{1-p}{d_j}\right)^2 \sum_{j' \sim j} \sum_{j'' \sim j} f(j', i) f(j'', i) \left[1 - \frac{d_j}{d_j - 1} \mathbb{I}(\text{do not share first edge})\right]. \quad (54)$$

This can be rewritten

$$p^2 \left(\frac{1-p}{d_j}\right)^2 \frac{d_j}{d_j - 1} \sum_{j' \sim j} \sum_{j'' \sim j} f(j', i) f(j'', i) \left[\mathbb{I}(\text{share first edge}) - \frac{1}{d_j}\right]$$

$$= p^2 (1-p)^2 \frac{1}{d_j - 1} \left\{\frac{1}{d_j} \sum_{j' \sim j} f(j', i)^2 - \left(\frac{1}{d_j} \sum_{j' \sim j} f(j', i)\right)^2\right\} \geq 0 \quad (55)$$

where we used Jensen's inequality.

To complete the proof, we also consider the subcase $i = j$. For i.i.d. walkers, the correlation term evaluates to

$$p^2 \left[\frac{1}{1 - (1-p)\mathbf{P}}\right]_{ii}^2 = p^2 \left[1 + \frac{(1-p)\mathbf{P}}{1 - (1-p)\mathbf{P}}\right]_{ii}^2$$

$$= p^2 \left(1 + 2(1-p) \left[\frac{\mathbf{P}}{1 - (1-p)\mathbf{P}}\right]_{ii} + (1-p)^2 \left[\frac{\mathbf{P}}{1 - (1-p)\mathbf{P}}\right]_{ii}^2\right) \quad (56)$$

For repelling walkers in the same block, we need to consider contributions from i) both walkers terminating immediately, ii) one terminating and one leaving then returning to $i$, and iii) both walkers leaving (to different neighbours ($i' \neq i''$) then returning to $i$. Enumerating these possibilities, we get:

$$p^2 \left(1 + 2(1-p) \left[\frac{\mathbf{P}}{1 - (1-p)\mathbf{P}}\right]_{ii} + \right.$$

$$\left. \left(\frac{1-p}{d_i}\right)^2 \frac{d_i}{d_i - 1} \sum_{i' \in \mathcal{N}(i)} \sum_{i'' \in \mathcal{N}(i) \setminus i'} \left[\frac{1}{1 - (1-p)\mathbf{P}}\right]_{i'i} \left[\frac{1}{1 - (1-p)\mathbf{P}}\right]_{i''i}\right). \quad (57)$$

Only the final term differs compared to Eq. 56. It is of precisely the same form as considered above with $i = j$, so we immediately deduce that it is smaller with repelling walkers in the same block. It follows that when $i = j$ repelling random walkers also yield lower variance estimators $\widehat{\boldsymbol{\pi}}_j$.

We have now considered the cases where the pair of walkers are in either different blocks or the same block, including the sub-cases $i \neq j$ and $i = j$, and proved that the variance of the estimator $\widehat{\boldsymbol{\pi}}$ is the same or reduced in both cases. The proof is complete. $\square$

### A.3    PROOF OF THEOREM 4.4 (VARIANCE OF STEP-BY-STEP LINEAR FUNCTIONS IS REDUCED BY TRANSIENT REPULSION)

In this section, we supplement the discussion at the end of 4, identifying a general class of functions whose variance is suppressed by conditioning that random walks exhibit transient repulsion.

Following Def. 4.3, let $\Omega_r$ denote the set of all infinite-length walks starting at node $r$, $\Omega_r := \{(v_i)_{i=0}^\infty \mid v_0 = r, v_i \in \mathcal{N}, (v_i, v_{i+1}) \in \mathcal{E}\}$. Recall that we refer to a function $y : \Omega_r \to \mathbb{R}$ as *step-by-step linear* if it takes the form:

$$y(\omega) = \sum_{i=0}^\infty f(v_i, i) \prod_{j=1}^i g(v_{j-1}, v_j, j, j-1). \tag{58}$$

An example of such a function provided by $\phi(i)_k$, the $k$-th component of the graph random feature corresponding to node $i$, for which $f(v_i, i) = \mathbb{I}(v_i = k)$ and $g(v_{j-1}, v_j, j, j-1) = \frac{w_{v_{j-1}, v_j} d_{v_{j-1}}}{1-p} \mathbb{I}(t_j > p)$. Here, $w_{v_{j-1}, v_j}$ is the weight of the edge $(v_{j-1}, v_j) \in \mathcal{E}$, $d_{v_{j-1}}$ is the degree of the node $v_{j-1}$ and $t_j \sim \text{Unif}(0, 1)$ is a termination random variable which controls whether the walk ends at the $j$th timestep. Another example is provided by the $k$th component of the PageRank vector estimator $\hat{\boldsymbol{\pi}}_k$, in which case $f(v_i, i) = \mathbb{I}(v_i = k)\mathbb{I}(t_j > p)$ and $g(v_{j-1}, v_j, j, j-1) = \mathbb{I}(t_{j-1} < p)$, ensuring that $y(w)$ evaluates to 1 if the walker terminates at node $k$ and is zero otherwise.

In Theorem 4.4, we asserted that, for the estimator $Y := \sum_{i=1}^m y(\omega_i)$ (where $\{\omega_i\}_{i=1}^m$ enumerates $m$ (infinite) walks on $\mathcal{G}$ and $y : \Omega_r \to \mathbb{R}$ is a step-by-step linear function),

$$\text{Var}(Y_{\text{repelling}}) \leq \text{Var}(Y_{\text{i.i.d.}}). \tag{59}$$

This is proved as follows. Begin by writing

$$y(\omega^{(k)}) = f(v_0, 0) + \sum_{i=1}^\infty f(v_i^{(k)}, i) \prod_{j=1}^i g(v_{j-1}^{(k)}, v_j^{(k)}, j, j-1) = f(v_0, 0) + h\left((v_i^{(k)})_{i=1}^\infty\right) \tag{60}$$

with $k = \{1, 2\}$. Note that we took $v_0^{(1)} = v_0^{(2)} = v_0$ since the walkers begin at the same node, and introduced the function $h$ to simplify notation.

As in Sec. A.2, a pair of walks may be either assigned to the same block or different blocks (see Fig. 1). We focus on the former case since in the latter they are i.i.d. and the variance of the estimator is trivially unchanged. This means that our walkers diverge at the first timestep, $v_1^{(1)} \neq v_1^{(2)}$. It follows from the definition of transient repulsion that the random variables $h((v_i^{(k)})_{i=1}^\infty)$ are conditionally independent given $(v_1^{(1)}, v_1^{(2)})$ since at this point the walkers stop repelling. Moreover, since in both cases the marginal distribution over $\omega$ is identical, for $\text{Var}(Y)$ to be reduced by repulsion we just require that

$$\mathbb{E}_{\text{i.i.d.}}\left[ h\left((v_i^{(1)})_{i=1}^\infty\right) h\left((v_i^{(2)})_{i=1}^\infty\right) \right] \overset{?}{\geq} \mathbb{E}_{\text{rep}}\left[ h\left((v_i^{(1)})_{i=1}^\infty\right) h\left((v_i^{(2)})_{i=1}^\infty\right) \right]. \tag{61}$$

In the i.i.d. scheme, $v_1^{(1)}$ and $v_1^{(2)}$ are independent and are uniformly distributed among the set of neighbours $\mathcal{N}(v_0)$, each with probability $1/d_0$. Meanwhile, in the repelling scheme, $(v_1^{(1)}, v_1^{(2)})$ is uniformly distributed among the set $\{(v_i, v_j) \mid v_i, v_j \in \mathcal{N}(v_0), v_i \neq v_j\}$ – i.e. all $d_0(d_0 - 1)$ possible pairs of distinct neighbours of node $v_0$. Then Eq. 61 evaluates to

$$\frac{1}{d_0^2} \sum_{\substack{v_1^{(1)} \in \mathcal{N}(v_0) \\ v_1^{(2)} \in \mathcal{N}(v_0)}} \mathbb{E}(h|v_1^{(1)})\mathbb{E}(h|v_1^{(2)}) - \frac{1}{d_0(d_0 - 1)} \sum_{\substack{v_1^{(1)} \in \mathcal{N}(v_0) \\ v_1^{(2)} \in \mathcal{N}(v_0) \setminus v_1^{(1)}}} \mathbb{E}(h|v_1^{(1)})\mathbb{E}(h|v_1^{(2)}) \overset{?}{\geq} 0 \tag{62}$$

where we used that $h^{(1,2)}$ are conditionally independent given $v_1^{(1,2)}$. Rearranging, this expression is nothing other than

$$\frac{1}{d_0 - 1} \text{Var}_{v_1 \in \mathcal{N}(v_0)}(\mathbb{E}(h|v_1)) \overset{?}{\geq} 0 \tag{63}$$

where the variance is being computed over the $d_0$ nodes that neighbour $v_0$. Eq. 63 trivially holds, confirming that $\text{Var}(Y_{\text{repelling}}) \leq \text{Var}(Y_{\text{i.i.d.}})$. $\square$

Note that Theorem 4.4 does not obviate the long proof in Sec. A.1: suppressing the variance of the GRF coordinate $\phi(i)_k$, $i, k \in \mathcal{N}$ is *not* sufficient to conclude that the variance of the dot product $\hat{\mathbf{K}}_{ij} = \phi(i)^\top \phi(j)$ is also reduced since now we need to consider correlations between $\phi(i)_{k_1}$ and $\phi(i)_{k_2}$ with $k_1 \neq k_2$. On the other hand, it does subsume Theorem 4.2 as a special case, though we keep this section in the manuscript for clarity of presentation.

