# OpenReview forum: "Repelling Random Walks"
_ICLR.cc/2024/Conference — ICLR 2024 poster_

### Official Review · Reviewer_vsCv · 2023-10-31

**Soundness:** 3 good
**Presentation:** 3 good
**Contribution:** 3 good
**Rating:** 6
**Confidence:** 3

**Summary:**

This work introduced a repelling mechanism among walkers in a graph when doing MC simulation. The proposed sampling mechanism is easy to understand. And intuitively it makes more sense than iid sampling by considering the graph topological property. Experiments are conducted on three graph-related tasks. Results also verified its better performance than the iid baseline

**Strengths:**

S1. A more vivid random walk mechanism with considering graph topological property

S2. experiments on three graph tasks to show the advantage of the proposed

S3. Solid theoretical analyses

**Weaknesses:**

W1. concern on the audience interest

W2. more interesting downstream applications are expected

**Questions:**

Overall, this is a good paper with both solid theoretical analyses and experiments on various tasks. However, some concerns are:

C1. Random walk is one of the important research topic in graph, the fundamental research is worthy of applause. Random walk related approaches have also be applied in downstream tasks in the real-world applications, such as graph embedding and community detection. However, the focus of this paper seems to be more fundamental. Random walk theoretical research usually fits better in venues like graph theory (lean more on mathematics) and computing theory (e.g., STOC). So I have concern on audience interest for ICLR.

C2. More real-world related applications are expected. The authors applied the proposed random walk mechanism in three applications. But the three seem to be more abstract than those driven by real-world applications or hot topics in the current research community. For example, graph kernel approximation covers one major category of approaches for graph embedding. PageRank vector approximation is also one of the fundamental problems for graph embedding and community detection. Graphlet detection is used in subgraph representation. Compared to the three in this paper, graph embedding, subgraph representation, and community detection may be closer to real-world scenario. In recent 10 years, deep learning approaches attract more attentions in almost every research field. If the fundamental research like this paper can show enhancement against recent so-called advanced methods or benefit the recent popular approaches, that would be more exciting.

=================

After reading the authors response where more examples were involved. The response address some of my concerns. But more real-world related applications are expected (C2 in my review) in this paper.

As a result, I increased my score.

---

> ### Author Response · Authors · 2023-11-12
> **Rebuttal -- thank you for the review**
>
> We thank the reviewer for reading the text. We are pleased that they recognise the solid theoretical contributions and fundamental nature of the work -- indeed, to our knowledge, this is *the first QMC scheme applied to improve the sampling of random walks on graphs by correlating trajectories*. Indeed, to our knowledge it is the second QMC scheme of any kind to be applied to random walks, and it substantially outperforms the first which just correlates walk lengths (see 'q-a-GRFs' in Sec. 3 -- https://arxiv.org/abs/2305.12470, NeurIPS 2023, spotlight). We address their concerns and questions in full below.
>
> 1. **Audience concern**: Sampling efficiently is a problem of key interest in machine learning. Indeed, at ICML 2023 the best paper prize was awarded to a paper addressing precisely this problem on graphs (https://doi.org/10.48550/arXiv.2305.05097, ICML 2023, best paper), though this scheme provides only asymptotic guarantees, does not preserve the walks' marginal distributions and has limited downstream evaluation. The equivalent literature on variance-reduction techniques when sampling in Euclidean space is much older and better-established (see https://doi.org/10.1017/S0962492913000044 for a seminal work) but it also continues to grow quickly -- e.g. the `Performers' efficient transformer architecture samples orthogonal vectors and this 'repulsion' is found to be key to estimating the attention kernel efficiently (https://arxiv.org/abs/2009.14794, ICLR 2021, oral presentation). Though our work has a strong theoretical component and we endeavour to support our empirical conclusions with proofs, it is still squarely aimed at the machine learning community for whom performing efficient sampling will always remain a problem of central interest.
>
>
> 2. **Experimental section**: We sincerely thank the reviewer for their comments, but respectfully suggest that our new method does indeed benefit recent approaches. For example, in Sec. 3 we consider the problem of kernel approximation using graph random features (GRFs). GRFs were introduced as a scalable mechanism to estimate graph kernels just a few months ago (https://arxiv.org/abs/2305.00156, ICML 2023, oral presentation), and our simple QMC modification can reduce the error by a factor of $2$ at close to no cost. It provides bigger variance savings than q-a-GRFs (another QMC scheme designed *specifically* for the GRF setting) by an order of magnitude (https://arxiv.org/abs/2305.12470, NeurIPS 2023, spotlight), and in Table 1 we see that better kernel estimation permits better performance in relevant downstream applications. This shows that our algorithm is useful for contemporary work considered of interest by the community. We then not only apply it to PageRank estimation (where we again see strong empirical gains and provide theoretical guarantees), but also the estimation of graphlet statistics -- a real-world problem that the reviewer rightly identifies as important for graph embedding and subgraph representation. Here we see gains of more than 10% where previous variance reduction schemes such as non-backtracking walks have made only a very small difference (see Fig. 6 of https://arxiv.org/pdf/1603.07504.pdf, where the improvement is barely discernable). These three successful applications build confidence that our simple scheme will continue to prove effective as novel ML algorithms using random walks are proposed. We suggest that the fundamental nature of some aspects of this work does not put it at odds with recent ML developments, but rather it makes the techniques more interesting and general, and will help them remain useful as ML continues to evolve.
>
> We again thank the reviewer for reading the manuscript and invite them to respond with any further questions. We believe that we have addressed their concerns and hope that they consider raising their score.

---

> > ### Comment · Reviewer_vsCv · 2023-11-19
> > **Revise the review score**
> >
> > Hi, Authors,
> >
> > Thank you very much for the response and involve some examples. The response address some of my concerns. But more real-world related applications are expected (C2 in my review) in this paper.
> >
> > Overall, I would increase my score.

---

### Official Review · Reviewer_KSvf · 2023-11-02

**Soundness:** 2 fair
**Presentation:** 2 fair
**Contribution:** 2 fair
**Rating:** 6
**Confidence:** 4

**Summary:**

This paper presents a novel quasi-Monte Carlo mechanism called repelling random walks. The authors demonstrate that the marginal transition probabilities of repelling random walks remain unchanged compared to standard random walks. In particular, the paper proves that the variance of approximate random walk probabilities is suppressed by simulating repelling random walks. The paper showcases the effectiveness of repelling random walks by applying them to three distinct tasks.

**Strengths:**

S1. The paper introduces a novel quasi-Monte Carlo mechanism, repelling random walks, aimed at enhancing graph-based sampling. This approach could potentially inspire further research in this field.

S2. The marginal transition probabilities of repelling random walks remain unchanged, while the variance of these walks is reduced.

**Weaknesses:**

W1. The advantage of using repelling random walks over standard random walks appears to be marginal. For instance, the reduction in approximation errors when estimating PageRank using both standard and repelling random walks as shown in Table 2 is relatively minor.

W2.  The validity of certain arguments is heavily dependent on specific assumptions. Take Theorem 4.2, for example: its accuracy hinges on the assumption that the count of random walks is less than the minimum node degree in the provided graph. Nonetheless, in a variety of real-world network structures, the minimum node degree stands at one, rendering repelling random walks virtually indistinguishable from standard random walks.

W3. The paper's presentation needs improvements. The current manuscript contains ambiguous sentences and unclear notations. For instance:
- Page 2: the notation $P^{(i)}$ requires clarification, as the paper defines $P$ but not $P^{(i)}$.
- Page 3: the notation $i_1$ and $\delta_{i_1}$ require clarifications.
- In the appendix on Page 20: the reasoning behind the statement "only walkers originating from the same node are correlated" requires additional explanation.

---
During the rebuttal phase, the authors and I engaged in detailed discussions regarding the novelty and contributions of the paper. I appreciate that the authors have effectively addressed the issue related to the minimum node degree assumption (i.e., W2). We also had a thorough discussion about the fundamental differences between Repelling Random Walk and Radar Push, a closely related work. The authors' responses were not only prompt but also convincing. Consequently, I have decided to revise my initial score from 5 to 6.

**Questions:**

Q1. The experimental setups presented in Table 2 lack clarity. The paper indicates that 1000 trials are conducted on each graph, with more than two repelling random walks simulated during each trial. Could you specify the exact number of standard and repelling random walks simulated for each graph?

Q2. In Table 2, could you please indicate the minimum node degree for each graph?

---

> ### Author Response · Authors · 2023-11-12
> **Rebuttal -- thank you for the review**
>
> We thank the reviewer for reading the manuscript. We address all their concerns and questions below, clarifying points of misunderstanding.
>
> 1. **Size of the gains from repelling random walks**: We respectfully disagree that the gains from our scheme are marginal. Earlier notable Quasi-Monte Carlo schemes (which received much attention, but cannot be applied to our setting) often provide improvements of just a few percent (e.g. https://arxiv.org/pdf/2301.13856.pdf, ICML 2023, oral presentation), but here we leverage the specific graph structure to do substantially better. In the kernel application with $16$ walkers the estimation error is halved compared to the i.i.d. scheme (https://arxiv.org/abs/2305.00156, ICML 2023, oral presentation). Our method gives a bigger variance-saving compared to q-a-GRFs, the previous algorithm designed specifically for this problem, by an order of magnitude (https://arxiv.org/abs/2305.12470, NeurIPS 2023, spotlight). This improvement translates to much better performance in downstream tasks; for kernel regression on the biggest graph, *the prediction error is reduced by a factor of 3 at close to no computational cost*. In the PageRank experiment the improvements are still substantial at 5-10%. Meanwhile, for graphlet statistic estimation, we again see gains of over 10% where previous attempts to reduce estimator variance using non-backtracking walks have made little difference (see Fig. 6 of https://arxiv.org/pdf/1603.07504.pdf, where the improvement is barely visible). To summarise, we have provided a general QMC scheme that substantially suppresses the variance of a broad class of estimators. The gains are much larger than other approaches in the literature and the implementation is trival and computationally cheap, essentially involving sampling neighbours without instead of with replacement.
>
> 2. **Theoretical assumptions**: The reviewer is correct that our theoretical guarantees formally rely on particular simplifying assumptions -- specifically, that the number of walkers is smaller than the minimum node degree (which is of course typically not the case) and the 'transient repelling' property (which is less effective than the 'full-repelling' scheme in practice). To prove theorems on graphs it is very standard to make these kinds of assumptions; as discussed in the introduction, this is why theoretical results for the dynamics of interacting walkers are typically restricted to very simple topologies and to single self-interacting walkers rather than ensembles (e.g. https://doi.org/10.1214/EJP.v19-2669, https://doi.org/10.1016/j.spa.2022.03.007). Our claim is not that all the assumptions hold, but rather that the theoretical insights from these limits might help explain the scheme's excellent empirical performance. Empirically, we observe that even when the minimum node degree is 1 repelling walkers substantially outperform the i.i.d. counterpart across a range of tasks (see Secs 3-5). To quote the manuscript: 'though we have made some restrictions for analytic tractability, we will empirically observe that the full repelling QMC scheme is effective in much broader settings... Extending the proof to these general cases is an exciting open problem'. We hope this clarifies the purpose and presentation of the assumptions in our theoretical results.
>
> 3. **Presentation**: We are sorry that some of the notation was unclear to the reviewer. $\boldsymbol{P}^{(i)}$ is the transition matrix of a particular walker, where $1 \leq i \leq m$ indexes the walkers present. In this independent case, its entries are given by Eq. 1. In the short section 'physical interpretation and entanglement', $i_1$ and $i_2$ refer to the starting nodes of two walkers and $j_1$ and $j_2$ refer to their ending positions after a transition. $\delta_{i_1 i_2}$ is the standard delta function, which evaluates to $1$ when $i_1 = i_2$ and $0$ otherwise. 'Only walkers originating from the same node are correlated' refers to the fact that we simulate an ensemble of repelling walkers out of each node, but walkers originating out of two different nodes are independent. We will clarify these important points in the manuscript. We thank the reviewer for flagging these points of confusion.
>
> 4. **Experimental setup in Table 2**: For every graph, we simulate $2$ walkers out of every node. This pair of walkers is either i.i.d. or repelling. We use these to construct an estimate of the PageRank vector, and measure its quality by comparing to the groundtruth. We repeat this $1000$ times and report the mean approximation error with the standard deviation of that mean. We have clarified this in the manuscript. Concerning the minimum node degree, please see our earlier comments in point 2.
>
> We again thank the reviewer for reading the manuscript. We believe we have addressed all their concerns and questions in full and warmly invite them to reply if anything remains unresolved. We hope they will consider raising their score.

---

> > ### Author Response · Authors · 2023-11-14
> > **Updated pdfs**
> >
> > The reviewer might be interested to inspect the updated main and supplementary pdfs, where changes are indicated in red. In particular, we have clarified the meaning of $\mathbf{P}^{(i)}$ and $i_1,i_2,j_1,j_2$, remarked that $\delta_{i_1 i_2}$ is the standard delta function, improved the explanation of the experimental setup for Table 2, and explained why walkers starting at different nodes are independent in our algorithm. We thank the reviewer for prompting us to make these changes and look forward to their response.

---

> > ### Comment · Reviewer_KSvf · 2023-11-17
> > **Response to the authors' rebuttal**
> >
> > Thank you for the quick response! However, my major concern, as described in W2, still remains. The paper's theoretical analyses heavily rely on some very strong assumptions, which significantly differ from real-world cases, as admitted in the response. Without these assumptions, evaluating the superiority of the repelling random walks becomes challenging. This issue undermines the paper's technical contributions. In addition, regarding the presentation issue, it would be beneficial to include an explanation in the manuscript of the index $i$ for $P^{(i)}$, similar to the one provided in the rebuttal box.

---

> ### Author Response · Authors · 2023-11-17
> **Further clarification**
>
> We thank the reviewer for their reply, and will clarify the meaning of the index $i$ in the text.
>
> Whilst we respectfully maintain that making simplifying assumptions is a standard feature of rigorous theoretical work (even *with* these assumptions the proofs still span 12 pages), we are happy to show how some might be relaxed.
>
> The reviewer is particularly concerned by the assumption that the number of walkers is smaller than the minimum node degree. We make this so that the repelling walkers diverge to different neighbours at the first timestep. Suppose that this is not true and we have e.g. $2$ repelling walkers beginning on a node of degree $1$. In this case, in both the i.i.d. and repelling schemes, both walkers will hop to the single neighbouring node at the first timestep. Unless we have a trivial graph of just two nodes and a single edge, this second node is then guaranteed to have more than $1$ neighbour. Therefore, the repelling walkers will now diverge at the *second* timestep rather than the first. This means that we still get the benefits of RRWs exploring the graph better; the 'better exploration' just begins a timestep later.
>
> To parse this into algebra: consider Sec. 4.2, which proves the superiority of RRWs for PageRank estimation. Suppose we have e.g. $m=2$ walkers but the minimum node degree is $1$. Our arguments up to Eq. 51 are unmodified: the variance still depends on the probability that two walkers starting at the same node $j$ terminate at the same node $i$, and we want to to suppress this probability. Contributions from nodes of degree $d(j)>m$ (which is often the case for almost every node) are unchanged -- we still get the same variance savings already reported in the manuscript. For the minority of terms where $d(j)=1$ ('leaves'), the repelling and i.i.d. walkers have the same behaviour for the first timestep, moving to the neighbouring node $k \coloneqq \mathcal{N}(j)$ or terminating with probability $p$. They then repel at the *second timestep*, which we can reflect in Eqs. 52 and 53 by replacing the indices $j$ with $k$. The rest of the argument is essentially unchanged.
>
> We refrained from including this discussion in the original manuscript for compactness (the theory is already very long without this extra case-by-case analysis), but on reflection we agree that it might interest the motivated reader. We will add this discussion to the manuscript, and sincerely thank the reviewer for this suggestion. We hope it will make the nature of this particular assumption clearer.
>
> We trust that this clarifies our theoretical assumptions: they help streamline the argument and in places help tractability, but do not modify the substance of the conclusions. We stress again that the scemes works very well in practice, in places outperforming the previous best algorithm by an order of magnitude.
>
> May we clarify anything else for the reviewer, or are they satisfied with the above?

---

> ### Comment · Reviewer_KSvf · 2023-11-19
> **Follow-up response to authors' rebuttal**
>
> Thank you for the reply. I concur with the authors that the full benefits of Repelling Random Walks may become more evident over time, particularly due to the presence of one-degree nodes. I recommend that the authors incorporate the justifications provided in their further clarification into future versions of this manuscript.
>
> On another note, I encountered a related work [RP, AAAI'19], which focuses on distributed PageRank computation and introduces an algorithm called Radar-Push (RP). The main idea of Repelling Random Walks aligns closely with that of the RP algorithm as proposed in [RP, AAAI'19]. Specifically, the RP algorithm generates $d(v)$ random walks from each node $v$, where $d(v)$ denotes the degree of node $v$ in an undirected graph $G$. In each walking step, the RP algorithm does not select a neighbor of $v$ uniformly at random to extend the walk. Instead, it manually assigns the $d(v)$ random walks as evenly as possible (see Page 3 in [RP, AAAI'19] for details). A sketch to illustrate how the RP algorithm works is given in Figure 1 in [RP, AAAI'19]. Lemma 1 in [RP, AAAI'19] proves the unbiasedness of the RP algorithm. Lemma 2 demonstrates the superiority of the RP algorithm over the standard i.i.d random walks.
>
> [RP, AAAI'19] Luo S. Distributed PageRank Computation: An Improved Theoretical Study[C]//Proceedings of the AAAI Conference on Artificial Intelligence. 2019, 33(01): 4496-4503.
>
> It is also noteworthy that the RP algorithm is presented in the context of distributed PageRank computation, where bottlenecks primarily occur at hub nodes in the graph structure. To reduce the likelihood of multiple walks visiting hub nodes simultaneously, the RP algorithm generates $d(u)$ random walks from each node $u$ in each round. Importantly, the overall analysis of RP does not depend on any specific degree assumptions.
>
> Further discussion is welcomed on whether there are any fundamental differences between this manuscript and [RP, AAAI'19].

---

> ### Author Response · Authors · 2023-11-20
> **Manuscript update and further response**
>
> We thank the reviewer for their further response.
>
> Following our previous message, we invite the reviewer to consider the updated version of the manuscript. As suggested, **we have now explicitly relaxed the assumption that the number of walkers is smaller than the minimum node degree for the PageRank estimator, and updated the proof in full (see p. 22)**. We trust that this will allay their concerns.
>
> Regarding their more recent message, we sincerely thank the reviewer for bringing Luo's excellent work to our attention, and agree that it bears some interesting resemblances to (as well as key differences from) our own.
>
> Luo constructs walks by starting with $d_i$ walkers at every node $i \in \mathcal{N}$, with $d_i$ the respective node degree. At every timestep they are assigned to the $d_i$ neighbours by a random permutation. This means that every edge is traversed by one walker, and therefore that the number of walkers at node $i$ remains equal $d_i$ throughout the process. **Permutation is a special case of sampling without replacement when the number of walkers to be assigned is equal to the number of neighbours, so initially this is a special case of RRW dynamics**. However, this construction imposes constraints when computing estimators. For example, degree-$1$ nodes are forced to simulate just one random walk so may be undersampled. Meanwhile, high-degree nodes must simulate very many walks which may become prohibitively expensive (e.g. for a large Erdős–Rényi graph with a fixed edge-generation probability). PageRank is the steady state of a random walk process where the starting node is chosen uniformly at random, so to construct an unbiased estimator Luo has to correct for their unbalanced samples using importance sampling (see bottom right of p4501 of Luo's paper). Meanwhile, with RRWs we can freely choose the number of walks out of each node, so a simple drop-in replacement for the existing independent random walk generator is possible. Also, Luo's scheme introduces correlations between walkers from all nodes, whereas in our setting only ensembles out of each particular node interact. This means that one only needs to keep track of walker positions for each particular ensemble rather than every single walker on the graph simultaneously, which may make our algorithm easier to distribute.
>
> The reviewer notes that 'RP does not depend on any specific degree assumptions'. They achieve this by assuming a very specific walker initialisation with $d_i$ walkers at every node. In a sense, this is even more restrictive than the assumptions we initially made in our theoretical work (now relaxed), where we started with $d_i$ *or fewer* walkers at every node. Constraining the minimum node degree and constraining the walker initialisation are different ways of satisfying the same requirement. Both can be used to avoid explicit analysis of the more challenging dynamics when we have more than $d_i$ walkers at a node of degree $d_i$ (again, now incorporated into our manuscript). In short, Luo does not make fundamentally different assumptions about the graphs considered.
>
> We also emphasise again that our contributions are more general than estimating PageRank alone, which constitutes but one possible application in the manuscript. We not only identify a broad class of functions called 'step-by-step linear' whose variance is suppressed by RRWs (with PageRank as a special case), but also leverage RRWs for the recent application of estimating graph kernels. Here the gains are much bigger and the theoretical analysis is harder (we care about the distribution over walkers' subwalks, not just the node they terminate at). We also consider the task of graphlet concentration estimation. Our paper is not just about improving estimates of PageRank, but rather a general-purpose QMC sceme for better sampling on graphs. Nonetheless, we thank the reviewer for their helpful pointer and have now rightly cited Luo's interesting work (see the updated file).
>
> We thank the reveiwer again for the continued engagement with the review process and hope that, now we have explicitly relaxed the node degree assumption (which as Luo shows is standard in the literature), they might again consider raising their score.

---

> > ### Comment · Reviewer_KSvf · 2023-11-23
> > **Replying to the authors' further response**
> >
> > Thanks for the reply. I am glad to see that the authors have successfully relaxed the assumption on the minimum node degree mentioned in the original manuscript. The explanations on the major differences between the Repelling Random Walks proposed in this paper and the Radar Push algorithm given by Luo in AAAI'19 are also very convincing. These two points addressed my major concerns regarding the paper's novelty and contributions. In light of these improvements, I would like to raise my rating from 5 to 6.

---

> > > ### Author Response · Authors · 2023-11-25
> > > **Thanks for raising the score**
> > >
> > > We warmly thank the reviewer for their reply and raising their score. Their thoughtful feedback has improved the paper.
> > >
> > > We are pleased to report that, following further work and some minor tweaks to the manuscript, we were able to relax the node degree assumption for estimators of the more general class of 'step-by-step linear' functions (Thm 4.4). Although our rebuttal period was extended we are currently unable to upload new pdfs, but once this is fixed in OpenReview we invite the reviewer to inspect the changes.
> > >
> > > We again thank the reviewer for their time.

---

### Official Review · Reviewer_1jyG · 2023-11-19

**Soundness:** 3 good
**Presentation:** 3 good
**Contribution:** 2 fair
**Rating:** 6
**Confidence:** 4

**Summary:**

In this paper, the authors present repelling random walks to sample from a graph. In some examples, theoretical results about the improvement on the concentration of estimators and numerical experiments about efficiency in sampling are given. Both theoretical and numerical results look sound.

**Strengths:**

S1. A novel quasi-Monte Carlo algorithm called repelling random walks is given.

S2. Results on typical examples are given to illustrate the new algorithm.

**Weaknesses:**

W1. Theoretical results only show that the new variance of estimator is less than classical method, but the author can give a more explicit quantitative analysis of how small it can be.

W2. As to the efficiency in sampling, only numerical results are given, which weaken the solidity of the improvement brought by the new algorithm.

**Questions:**

Q1. Is it possible to give a more explicit relationship comparing the variances of estimators between the classical and the new method?

Q2. Ideally I would like to see some basic properties or results concerning the repelling random walk in Section 2 before diving into the applications.

---

> ### Author Response · Authors · 2023-11-20
> **Rebuttal -- thank you for the review**
>
> We sincerely thank the reviewer for agreeing to check our paper at short notice. We are pleased that they note the sound theoretical and experimental results and the novelty of the algorithm. We address their questions and concerns in detail below.
>
> 1. **Quantitative analysis of the estimator variance**: We thank the reviewer for their interesting question. We do actually give explicit forms for the variance of various estimators in the appendix (see e.g. Eqs. 32-34 and 52-57), but they are typically very complicated functions of the weighted adjacency matrix and difficult to reason about. This is why we resort to simply considering their sign. The reviewer is right to identify studying the derived closed forms in more detail as an important direction for future work, but with $10$ pages of algebra to justify the *presence* of an improvement, let alone reason about its behaviour depending on e.g. the graph topology, we must defer this challenging question to the future. The strong empirical results also give us confidence that the size of the gain provided by RRWs is big and applies to many graphs and estimators.
> 2. **Graphlet statistics**: The reviewer is correct that providing theoretical guarantees for the graphlet statistics estimator is difficult. Proving the mixing properties of these types of modified random walks is of central interest in pure mathematics (see e.g. https://arxiv.org/pdf/math/0610550.pdf, Communications in Contemporary Mathematics 2007, which considers the non-backtracking walk). It would be interesting to obtain similar guarantees for our scheme in the future, but this is likely to involve substantial work. This research may also be more suitable for a different venue.
> 3. **Basic properties of the RRW**: We thank the reviewer for their comment, but are unsure what further properties they would like to see discussed in Sec. 2. We currently present the algorithm (Def. 2.1) then explore some basic features: the unmodified marginal walk distribution (so estimators stay unbiased), the computational cost and the algorithm's implementation, and a physical interpretation in terms of mutual information and quantum entanglement that may help build intuition. Next we move on to give 3 examples of estimators whose concentration properties are improved, providing theoretical guarantees where possible. If the reviewer could kindly clarify what further discussion would improve readability we would be very happy to consider modifying the manuscript.
>
> We again thank the reviewer again for their time and comments. If satisfied with the above, we hope they might consider raising their score.

---

> > ### Author Response · Authors · 2023-11-25
> > **Any further questions?**
> >
> > May we clarify anything else for the reviewer or are they satisfied with the response above? If so, we respectfully hope that they might consider raising their score.

---

### Meta-Review · Area_Chair_iJoS · 2023-12-08

**Metareview:**

The paper discusses a new method for sampling from a graph, called "repelling random walks". This approach is shown to improve upon standard random walk methods. It keeps the transition probabilities similar to regular random walks, but reduces the variance in the estimates produced. The method is practical and easy to understand, taking into account the graph's structure. Its effectiveness is demonstrated through experiments on three graph-based tasks, where it outperforms traditional independent and identically distributed (iid) sampling methods. The paper provides both theoretical and numerical evidence to support these claims.

**Justification For Why Not Higher Score:**

Although all reviewers agreed on accepting the paper, the discussions and feedback indicated that the level of novelty in the content did not justify a recommendation for a "spotlight" presentation.

**Justification For Why Not Lower Score:**

The manuscript provides valuable new insights supported by robust numerical research. Its acceptance will be beneficial to the community.

---

### Decision · Program_Chairs · 2024-01-16

Accept (poster)